## RESEARCH REPORT

# TMO5 regulates PIN1 polarity convergence and organogenesis downstream of MONOPTEROS in the *Arabidopsis* shoot

Abdul Kareem*, Carolyn Ohno and Marcus G. Heisler‡

## ABSTRACT

Plants continuously produce lateral organs, such as leaves and flowers, from the shoot apical meristem (SAM). This process is guided by the accumulation of the plant hormone auxin and the polar localization of the efflux protein PIN-FORMED1 (PIN1). The transcription factor MONOPTEROS (MP) plays a crucial role in orienting PIN1 polarity, thereby facilitating auxin-driven organogenesis. In this study, we investigate genes downstream of MP that may regulate PIN1 polarity and organogenesis, discovering that the downstream vascular transcription factor TMO5 can promote PIN1 polarity convergence non-cell-autonomously and that TMO5 and its family members promote organ initiation in the SAM. By examining the role of auxin and cytokinin downstream of these genes, we provide evidence that the TMO5-like genes control PIN1 polarity and drive organogenesis by coordinating multiple hormonal signalling pathways.

KEY WORDS: TMO5, Auxin, Cytokinin, Plant, Organogenesis

## INTRODUCTION

Plants generate lateral organs such as leaves and flowers periodically from the periphery of the shoot apex or shoot apical meristem (SAM). While the resulting patterns, or phyllotactic arrangements of organs, have fascinated biologists and artists alike for hundreds of years (Jean and Barab, 1998), our understanding of the underlying molecular mechanisms remains far from complete. Organ outgrowth is triggered by local accumulation of the plant hormone auxin, which is transported to specific locations within the SAM periphery via the polarly localized membrane-bound efflux carrier protein PIN-FORMED1 (PIN1) (Heisler et al., 2005; Okada et al., 1991; Reinhardt et al., 2000, 2003). Studies have now demonstrated that the patterns of PIN1 polarity required to concentrate auxin at organ initiation sites occur due to positive feedback between auxin signalling and the polarity of PIN1 localization in neighbouring cells (Bhatia et al., 2016). This feedback loop is mediated by the MONOPTEROS (MP) [also known as AUXIN RESPONSE FACTOR 5 (ARF5)] transcription factor (Hardtke and Berleth, 1998; Przemeck et al., 1996), which gradually orients PIN1 polarity non-cell autonomously

School of Life and Environmental Sciences, University of Sydney, Sydney, NSW 2006, Australia.
*Present address: Department of Plant Biology, Linnean Center for Plant Biology, Swedish University of Agricultural Sciences, Almas allé 5, 756 51 Uppsala, Sweden.

‡Author for correspondence (marcus.heisler@sydney.edu.au)

A.K., 0000-0002-0976-8652; M.G.H., 0000-0001-5644-8398

towards MP-expressing cells, perhaps by altering mechanical stresses (Bhatia et al., 2016). Understanding how MP activity induces changes to PIN1 polarity remains a central priority for the field.

Previous work has identified several downstream targets of MP (Cole et al., 2009; Konishi et al., 2015; Schlereth et al., 2010; Yamaguchi et al., 2013). Some of these targets, including LEAFY (LFY), AINTEGUMENTA (ANT) and AINTEGUMENTA-LIKE6 (AIL6), are known to contribute to flower development. However, the degree of rescue of the flowerless *mp* phenotype by LFY and ANT expression is limited (Yamaguchi et al., 2013). Other known MP targets, including Target of MONOPTEROS 5 (TMO5) and Target of MONOPTEROS 6 (TMO6), act downstream of MP in vascular tissue and embryonic root development (Schlereth et al., 2010). For example, the basic Helix-Loop-Helix (bHLH) transcription factor TMO5 promotes periclinal divisions in developing root vasculature through a cytokinin biosynthesis pathway (De Rybel et al., 2014, 2013; Ohashi-Ito et al., 2014). However, when mis-expressed with its dimerization partner LONESOME HIGHWAY (LHW), TMO5 triggers ectopic leaf initiation on the petiole (De Rybel et al., 2014). Furthermore, TMO5 and its close relatives are expressed in the shoot where new organs initiate (Ram et al., 2020; Mor et al., 2022), suggesting that TMO5 and potentially other vascular-expressed MP targets may function in organogenesis.

Here, we investigate the pathway downstream of MP in lateral organ formation, by assessing the role of several genes already known to be MP targets. We demonstrate that, like MP, the downstream target TMO5, can regulate PIN1 polarity in the SAM epidermis non-cell autonomously and promote organogenesis. We also present evidence that TMO5 and its close homologues promote organogenesis by amplifying and triggering auxin and cytokinin signalling, respectively.

## RESULTS AND DISCUSSION

To identify genes downstream of MP involved in regulating organ initiation and PIN1 polarity in the shoot, we investigated several genes known to be regulated by MP and expressed in the shoot including *TMO5* (De Rybel et al., 2014, 2013; Ram et al., 2020; Mor et al., 2022), *TMO6* (Miyashima et al., 2019; Smet et al., 2019; Ram et al., 2020) and *ATDOF5.8* (Konishi et al., 2015; Larrieu et al., 2022). First, we assessed the expression of these genes relative to the *PIN1*, which marks organ initiation sites (Heisler et al., 2005). Using Yellow fluorescent Protein for Energy Transfer (YPET), we generated TMO5::TMO5-2YPET, TMO6::TMO6-2YPET and DOF5.8::DOF5.8-YPET reporters and transformed these constructs into plants already expressing PIN1 fused to Cyan Fluorescent Protein (CFP). In both vegetative and inflorescence meristems, expression of both TMO5::TMO5-2YPET and DOF5.8::DOF5.8-YPET was detected in epidermal and sub-epidermal layers of organ primordia coinciding with PIN1-CFP expression at position I1 when PIN1-CFP expression first appears sub-epidermally before PIN1 reversal in the epidermis (Heisler et al., 2005) (Fig. 1A,B; Fig. S1A,B,D). In contrast,

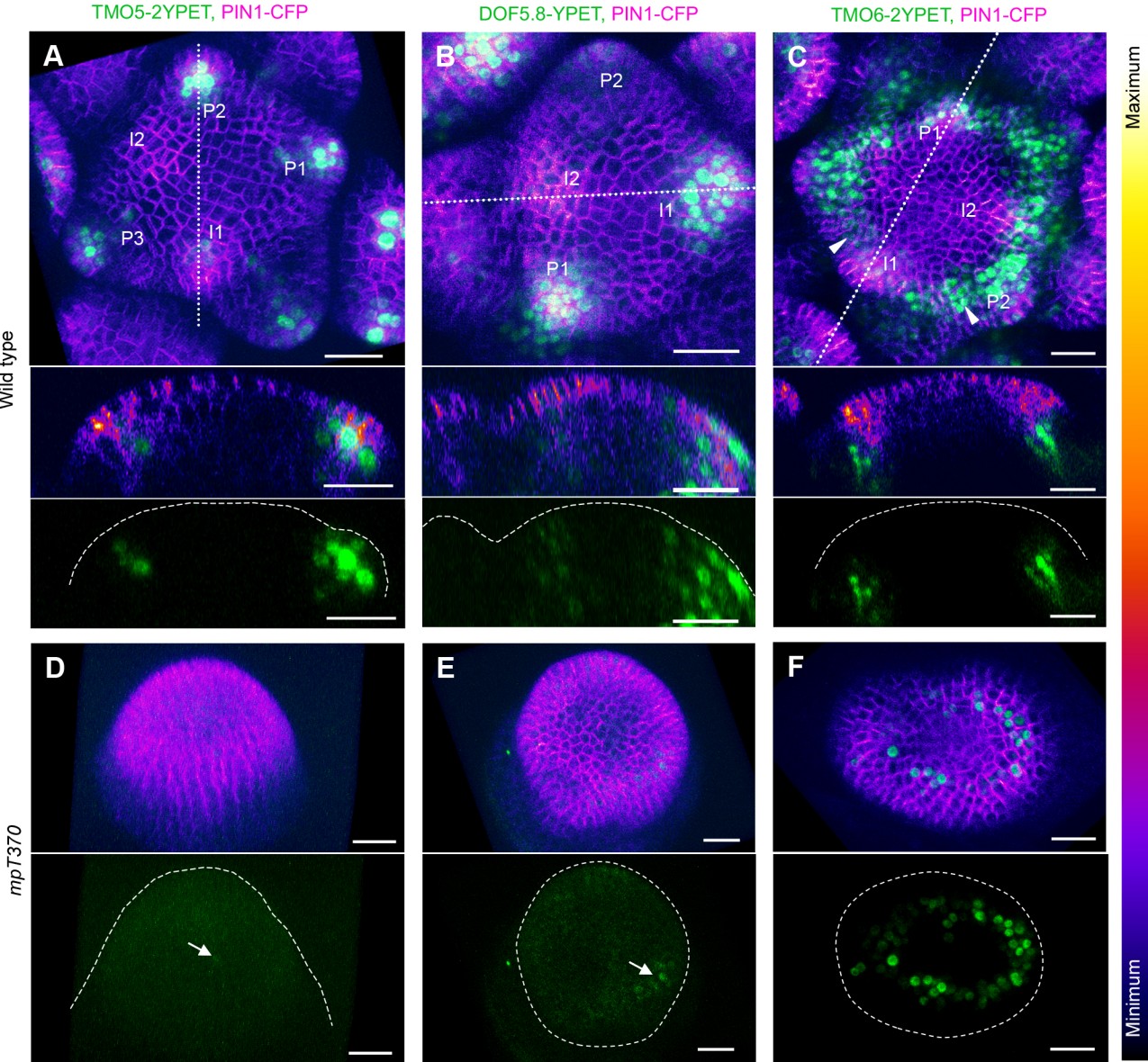

**Fig. 1. Expression patterns of TMO5, DOF5.8 and TMO6 in wild-type and *mp* mutant meristems.** (A) TMO5::TMO5-2YPET expression (green) in wild-type inflorescence meristem, partially overlaps with PIN1::PIN1-CFP expression (intensity-based colouring; the look-up table displays the signal intensity range) (*n*=13). (B) DOF5.8::DOF5.8-YPET expression (green) in wild-type inflorescence meristem, overlaps with PIN1::PIN1-CFP signal (*n*=12). (C) TMO6::TMO6-2YPET expression (green) in wild-type inflorescence meristem (*n*=14). Lower panels in A-C show corresponding longitudinal optical sections with both channels (top) and green channel alone (bottom). Incipient primordia (I1, I2) and developing primordia (P1, P2, P3) are labelled. Arrowheads in C indicate TMO6 expression between primordia. (D-F) TMO5, DOF5.8 and TMO6 expression in the *mp*-T370 mutant dome meristem, with lower panels displaying the green channel (*n*=15, 10, 11 from left to right). Dotted straight lines in A-C indicate the optical section planes; dashed outlines mark meristem boundaries in the green channel images of A-C and D-F. Barely detectable TMO5 and DOF5.8 signals are marked with arrows. Scale bars: 20 μm.

TMO6::TMO6-2YPET expression was not only detected at primordium positions but also, to a lesser extent, throughout the periphery, although this expression was only visible in sub-epidermal and inner cells (Fig. 1C; Fig. S1F). To test whether the expression of these genes depends on MP, we assessed our reporters in *mp* mutant meristems. We found almost no detectable TMO5 expression in the strong *mp* mutant (*mp-T370*) dome meristem or pin-like inflorescence meristem (Fig. 1D). However, TMO5 expression was detected in the rarely formed *mp* mutant leaves (Fig. S1C). Similarly, DOF5.8:: DOF5.8-YPET expression was low in the *mp* pin-like inflorescence meristem but was detected in leaf primordia that had developed from the vegetative meristem (Fig. 1E; Fig. S1E). TMO6::TMO6-2YPET expression was still detected in the *mp* mutant pin-like meristem but

the expression domain was reduced in size compared to the wild type (Fig. 1F; Fig. S1G). These data confirm that TMO5 and DOF5.8 are at primordium positions marked by high PIN1 expression mainly in sub-epidermal cells, from stage I1, while TMO6 is expressed more broadly.

We next tested whether localized expression of these genes in the *mp* mutant meristem could promote organ formation and alter PIN1 localization patterns. Towards this end, we generated small clones of cells expressing these genes individually in the *mp* mutant shoot meristem (*mp-T370* and *mp-B4149*, both strong alleles used to exclude ecotype-specific effects), harbouring a PIN1-CFP (or PIN1-GFP) marker. This was accomplished using an inducible Cre-lox recombination-mediated genetic mosaic system targeted to the

CLAVATA3 (CLV3) expression domain (pCLV3:CRE-GR+pUBQ10::GENE-2YPET), as described previously (Bhatia et al., 2016). We found the appearance of clones expressing TMO5, DOF5.8 or TMO6 in the *mp* dome-shaped shoot meristem within 2 days of DEX induction of Cre-GR. One to three days later,

PIN1 expression levels increased, within the clones expressing TMO5 but not DOF5.8 or TMO6 (Fig. S2A-C,E-H). Clonal induction of TMO5, but not DOF5.8 or TMO6, also led to organ outgrowth within 6-7 days of induction, resulting in the formation of flower-like structures (Fig. 2A-J; Fig. S2D). Transient ubiquitous

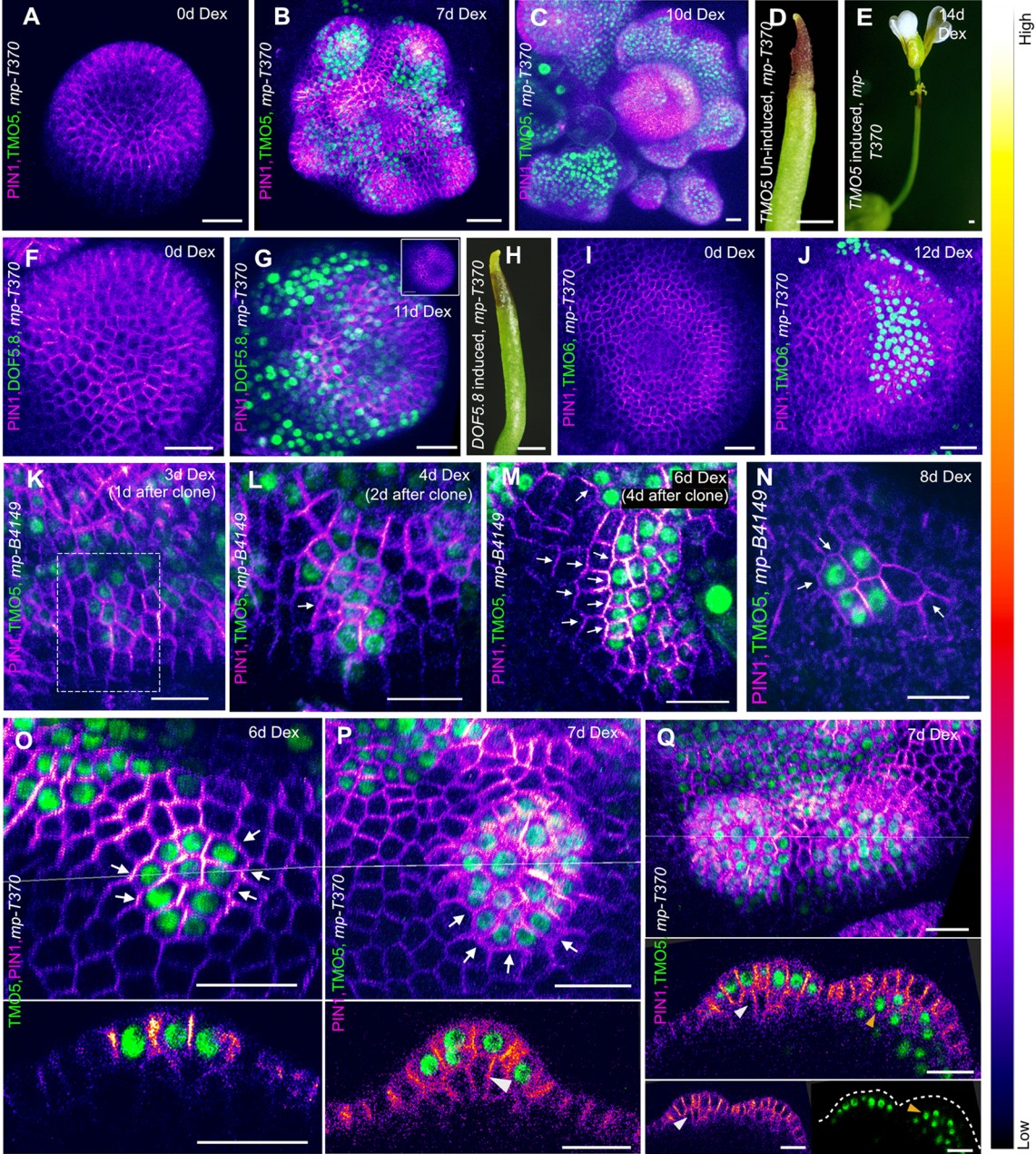

**Fig. 2. TMO5 promotes PIN1 polarity convergence and organogenesis independent of MP.** (A-C) Activation of PIN1-CFP expression (intensity-based colouring; the look-up table displays the signal intensity range), convergence formation and organ outgrowth in *mp* mutant shoot meristems after clonal activation of TMO5-2YPET (green). Panels A and B show the same meristem over time, whereas panel C presents a representative meristem (*n*=27). (D,E) Flower formation on *mp* pin-like inflorescence meristem following clonal activation of TMO5-2YPET (*n*=10). (F-J) No effect on PIN1 localization or organ outgrowth in *mp* mutant meristems with local activation of DOF5.8-YPET (F-H; *n*=12) or TMO6-YPET (I,J; *n*=15). Inset in G shows the meristem with the PIN1-CFP channel alone for clearer visualization. (K-M) Time-lapse images of the same *mp* mutant meristem region from 1 to 6 days after TMO5 clone formation, showing PIN1-GFP polarization in neighbouring cells towards the clones. The dashed box in K marks the area shown in L and M. Arrows indicate polarity (*n*=13). (N) PIN1-GFP polarization towards a TMO5 clone lacking organ outgrowth (*n*=9). (O,P) PIN1-CFP convergence in epidermal TMO5-expressing clones and polarized PIN1 in neighbouring cells in the *mp* mutant meristem at two time points of the same clones. Lower panels show longitudinal sections. Arrows indicate polarity; arrowhead in P marks sub-epidermal PIN1 expression (*n*=24). (Q) Two distinct organ outgrowths with epidermal PIN1 convergence induced by epidermal (left) or sub-epidermal (right) TMO5 clones. Lower panels show longitudinal sections. White arrowhead marks PIN1 activation in sub-epidermal cells; orange arrowhead marks sub-epidermal TMO5 clones. Dashed line marks the outgrowth boundary (*n*=5). Scale bars: 20 μm (A-C,F,G,I-Q); 1 mm (D,E,H).

expression of TMO5 also induced flower formation in *mp* mutants (Fig. S2I), while local clonal activation in wild-type plants resulted in altered organ positioning (Fig. S2J,K). Examining TMO5 clones more closely, we found that, similar to MP (Bhatia et al., 2016), localized TMO5 activity was sufficient to induce a localized PIN1 polarity convergence 1-2 days after clone appearance (Fig. 2K-M,O,P). PIN1 localization became polarized towards the TMO5 clones in cells adjacent to the clones, demonstrating a non-cell autonomous influence (Fig. 2L-P; Fig. S2L-O). This PIN1 polarity response was also observed from very small clones when organ growth never eventuated, indicating that PIN1 localization towards the clones is not necessarily associated with organ formation (Fig. 2N). Additionally, we observed activation of PIN1 expression in sub-epidermal cells located in layer 2 (L2) in response to overlying TMO5 clones in the epidermis, although the polarity of PIN1-GFP was not discernible (Fig. 2P,Q). We also detected upregulation of PIN1 expression and polarity convergence in the epidermal cells in response to underlying TMO5 clones in sub-epidermal L2 cells (Fig. 2Q). These sub-epidermal TMO5 clones were sufficient to induce organ outgrowth in the *mp* pin-like meristem, similar to the effect of epidermal clones (Fig. 2Q). Altogether, these data support the proposal that TMO5, but not TMO6 or DOF5.8, act downstream of MP to promote organ outgrowth and PIN1 polarity convergence non-cell autonomously.

Although changes to shoot meristem size have been reported for some mutant combinations of TMO5-like genes previously, no changes to organogenesis were reported (Mor et al., 2022). To investigate whether TMO5 contributes to organ formation, we started by analyzing organogenesis in *tmo5* single mutants but found no defects (Fig. S3B) (De Rybel et al., 2013; Vera-Sirera et al., 2015). Therefore, we investigated the function of genes closely related to *TMO5* including *TMO5-LIKE1* (*T5L1*) *T5L2*, *T5L3* (De Rybel et al., 2013), as well as a newly identified homologue, *AT2G40200*, which we named *T5L4* (Fig. S3A). Like other homologues, T5L4 has been reported to dimerize with the partnering protein LHW *in vivo* (De Rybel et al., 2013). We detected the expression of T5L1 in vascular tissue below the shoot and T5L2 and T5L3 in the shoot apical meristem, consistent with a previous report (Mor et al., 2022). We also found that *T5L4* was expressed in the meristem, albeit in a distinct pattern that appears to border primordium positions (Fig. 3A-D). As single mutants *t5l1*, *t5l2*, *t5l3* and *t5l4* did not display any defects in organogenesis (De Rybel et al., 2013; Vera-Sirera et al., 2015) (Fig. S3B), we investigated higher order mutants. While the *tmo5,t5l2,t5l3* triple mutant displayed a significant reduction in the number of flowers formed in the inflorescence compared to wild type, the *tmo5,t5l4* double mutant and *tmo5,t5l1,t5l4* triple mutant showed a greater reduction in the number of flowers formed (Fig. 3E). The *tmo5,t5l1,t5l2,t5l3* quadruple mutant showed the most severe phenotype and produced no flowers (Fig. 3E-I). Though the quadruple mutant germinated similarly to the wild type, it displayed retarded root and shoot development and a tiny stature (Fig. 3F,G). Shoot growth terminated after the 6-8 rosette leaf stage without producing an inflorescence (Fig. 3H,I). Phyllotaxis analysis of *tmo5,t5l1,t5l4* and *tmo5,t5l2,t5l3* triple mutants showed abnormal pattern of siliques (Fig. S3C) indicating that TMO5 and T5Ls contribute to organ positioning. To investigate whether this contribution depends on auxin, we treated *tmo5* mutants with 1 µM N-1-naphthylphthalamic acid (NPA), a concentration of the auxin transport inhibitor (Abas et al., 2021) that does not cause organogenesis defects in the wild type (Fig. 3J,N; Fig. S3D). Under such conditions we found that *tmo5* single

mutants produced a leafless meristem structure after making the first pair of leaves (Fig. 3K,L). These plants continued to grow and form a pin-like inflorescence (Fig. S3E,F). The *t5l4* single, *tmo5,t5l1* and *tmo5,t5l4* double mutants and *tmo5,t5l1, t5l4* triple mutant also produced pin-like inflorescences upon 1 µM NPA treatment (Fig. 3M,O; Fig. S3G-I). These data demonstrate that, despite the noted differences in expression, *TMO5* and the T5L genes contribute redundantly to organogenesis independently of shoot growth, in synergy with auxin and its polar transport.

To better understand how the organogenic activity of TMO5 relates to auxin, we perturbed auxin transport during TMO5 clone-induced organ outgrowth. We found that TMO5 induction was still able to induce organ formation in the presence of 10 µM NPA, which is a concentration of NPA sufficient to cause a pin-like inflorescence in wild type and suppress organ outgrowth in the *mp* mutant (Reinhardt et al., 2000; Schuetz et al., 2008), albeit with delayed and reduced frequency (Fig. 3P-S). In this case, polarization in response to the clones was harder to detect for large clones but still clearly visible for small clones (Fig. 3T). However, when auxin signalling (not transport) was reduced by treating with 100 µM auxinole, an inhibitor of auxin signalling, TMO5-induced organ formation was completely suppressed and no polarity response was detected (Fig. S3J,K), revealing that TMO5 clones require residual auxin signalling in the *mp* mutant to induce organ formation and polarize PIN1.

While our data show that TMO5-induced organogenesis in the *mp* mutant background requires residual auxin signalling, previous work has also shown that auxin alone is not sufficient to induce organ formation from *mp* mutant meristems (Reinhardt et al., 2003), indicating that other pathways must also be activated by TMO5 in order to induce organ growth. In addition to auxin, TMO5-LHW heterodimers are known to regulate cytokinin biosynthesis and signalling to induce periclinal cell divisions during vascular development (De Rybel et al., 2013; Ohashi-Ito et al., 2014). We therefore tested whether the promotion of organogenesis by induced TMO5 expression in the *mp* mutant meristem could be phenocopied by exogenous applications of cytokinin. Indeed, we found that external application of cytokinin with 10 µM trans-zeatin (TZ) could trigger growth from the *mp* vegetative meristem (Fig. 4A-C). However, cytokinin application alone did not initiate outgrowth from the *mp* pin-like inflorescence meristem. Instead, we found that a combined application of auxin [5 mM indole-3-acetic acid (IAA)] and a concentration of cytokinin (1 mM TZ) previously shown to induce outgrowth on dark-grown pin-like inflorescences in tomato (Yoshida et al., 2011), was sufficient to promote organ primordia-like outgrowth (Fig. 4D). All together, these data indicate that MP functions to promote organogenesis by regulating both auxin- and cytokinin-specific pathways, likely via TMO5.

Given our data imply that TMO5 acts downstream of MP to promote organogenesis by promoting both auxin and cytokinin signalling, we next tested whether auxin and cytokinin treatment can promote organ outgrowth in the *tmo5,t5l1,t5l2,t5l3* quadruple mutant shoot. We found that the one-off cytokinin application with 10 µM TZ at shoot tip promoted new leaf formation in the 5-day-old quadruple mutant and thus restored, to a degree, the vegetative defect (Fig. 4E,G,H; Fig. S4A). In contrast, application of 1 µM 1-naphthaleneacetic acid (NAA), a more penetrant form of auxin, did not promote leaf formation in the mutant (Fig. 4E,I; Fig. S4A), indicating that cytokinin is the primary limiting factor in the *tmo5, t5l1,t5l2,t5l3* mutant during vegetative growth. Although a one-off cytokinin application also led to inflorescence formation, few flowers

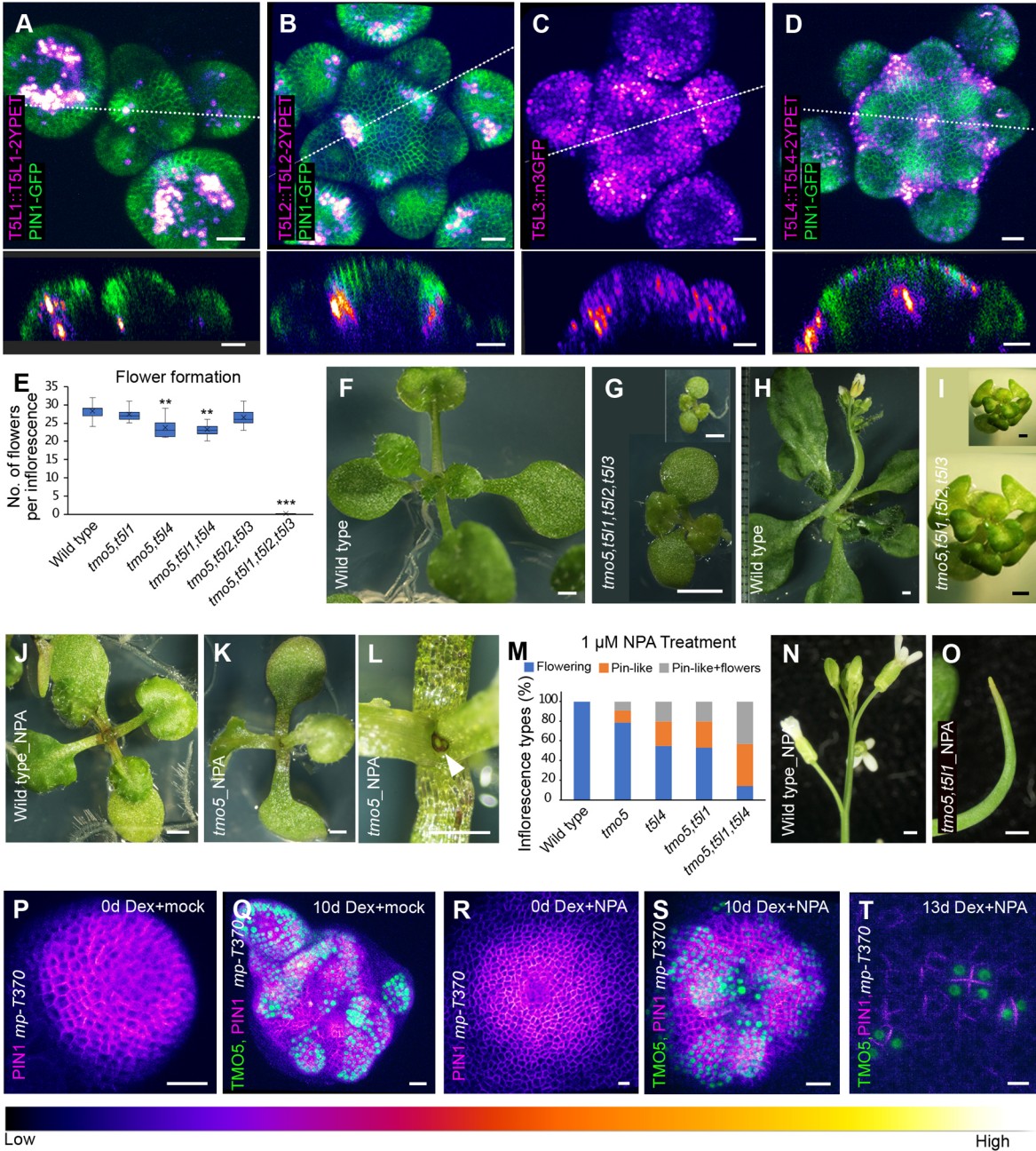

**Fig. 3. *TMO5* and TMO5-LIKE genes contribute to organogenesis.** (A-D) Expression patterns of T5L1, T5L2, T5L3 and T5L4 reporters (intensity-based colouring; the look-up table at the bottom displays the signal intensity range) in the wild-type inflorescence meristem with or without PIN1-GFP (green). Lower panels show longitudinal optical sections corresponding to the lines in the upper panels (*n*=10, 12, 12, 13 from left to right). (E) Average number of flowers formed per inflorescence in wild type and *tmo5/t5l* mutants. *n*=20 plants per genotype; **P<0.01; ***P<0.001 (Poisson regression). Box plot shows median (central line), mean (cross), first and third quartiles (top and bottom edges), and minimum and maximum values excluding outliers (whiskers). (F-I) Growth phenotypes of the homozygous *tmo5* quadruple mutant (*tmo5,t5l1,t5l2,t5l3*) at vegetative and reproductive stages compared to wild type. Insets in G and I show the mutant at the same scale as the corresponding wild type. Brightness and contrast adjusted in I. (J) Wild-type seedling after 8 days of 1 μM NPA treatment. (K,L) Leafless dome structure in a *tmo5* mutant under 1 μM NPA, with L showing a magnified view. (M) Frequency of formation of inflorescence types (flowering, pin-like and pin-like with flowers) in wild type and *tmo5/t5l* mutants following treatment with 1 μM NPA (*n*=15-35 plants per genotype). (N,O) Inflorescences of wild type and *tmo5,t5l1* mutants after 1 μM NPA treatment, showing normal flowering in wild type and pin-like inflorescences in the mutants. (P,Q) Mock treated *mp* mutant meristem at 0 days and 10 days following TMO5-2YPET (green) clonal activation showing organ formation with PIN1-CFP expression (intensity-based colouring) (*n*=12). (R,S) *mp* mutant meristem treated with 10 μM NPA for 0 days and 10 days after TMO5-2YPET clonal activation showing organ formation with PIN1-CFP expression (*n*=15). (T) Polarized PIN1 expression between cells within TMO5 clones in *mp* mutant treated with 10 μM NPA (*n*=5). Scale bars: 20 μm (A-D,P-T); 1 mm (F-L,N,O).

formed on the inflorescence (Fig. 4H). Therefore, we applied NAA to the cytokinin-pre-treated quadruple mutants to test whether cytokinin together with auxin can rescue flower formation in the quadruple mutant. Indeed, we found that auxin treatment enhanced flower formation (Fig. 4F,J). In contrast, the sole application of auxin was insufficient to promote leaf or flower formation in the quadruple

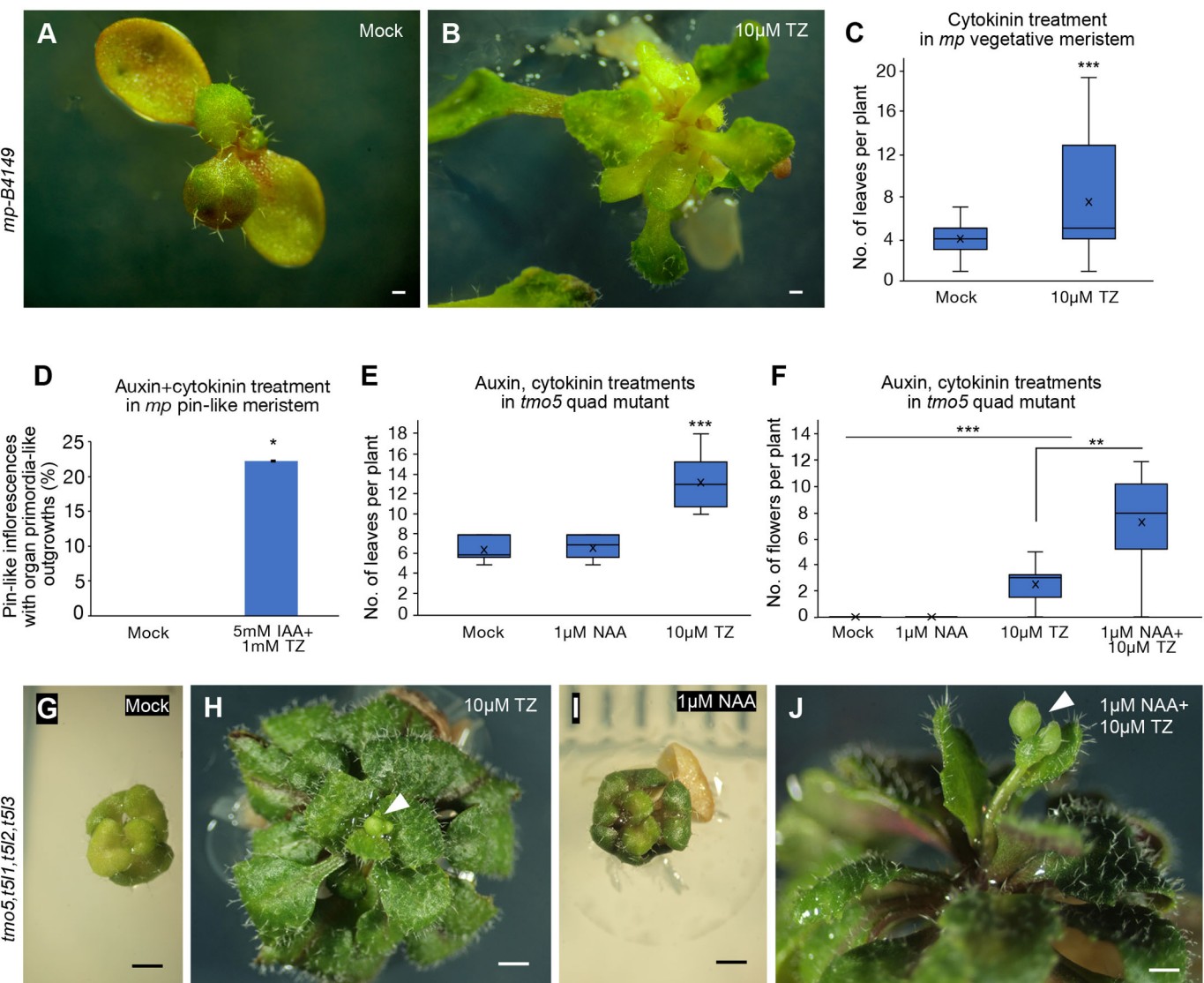

**Fig. 4. Cytokinin and auxin partially rescue TMO5/TMO5-LIKE function for organ formation.** (A,B) Leaf formation in *mp* mutant following 13 days of cytokinin treatment with 10 µM TZ (B) compared with mock treatment (A). (C) Average number of leaves per plant in the *mp* mutant following 10 µM TZ treatment [*n*=15 (mock) and 10 (TZ) plants]. ****P*<0.001 (Poisson regression). (D) Frequency of *mp* pin-like inflorescence meristems exhibiting organ primordia-like outgrowths after treatment with both auxin (5 mM IAA) and cytokinin (1 mM TZ) (*n*=18 plants per treatment). Data are mean±s.e.m. **P*<0.05 (two-tailed Student's *t*-test). (E) Average number of leaves per plant in the *tmo5,t5l1,t5l2,t5l3* quadruple mutant following auxin (1 µM NAA) and cytokinin (10 µM TZ) treatment (*n*=10 plants per treatment). ****P*<0.001 (Poisson regression). (F) Average number of flowers formed per plant in *tmo5,t5l1,t5l2,t5l3* quadruple mutants after auxin (1 µM NAA) and cytokinin (10 µM TZ) treatment (*n*=10 plants per treatment). ***P*<0.01; ****P*<0.001 (two-tailed Student's *t*-test). Box plots show median (central line), mean (cross), first and third quartiles (top and bottom edges), and minimum and maximum values excluding outliers (whiskers). (G-J) Leaf and flower formation in the *tmo5,t5l1,t5l2,t5l3* quadruple mutant after cytokinin and auxin treatment. Single 10 µM TZ application partially rescues leaf formation and induces flowering (H); 1 µM NAA alone has no effect (I) but enhances flowering following cytokinin application (J). Arrowheads in H and J indicate flower buds. Scale bars: 1 mm.

mutant (Fig. 4E,F). These data reveal that auxin and cytokinin rescue the organogenesis defects of *mp* and *tmo5,t5l1,t5l2,t5l3* mutants similarly. In both genetic backgrounds, a lack of cytokinin limits organogenesis during vegetative growth while a lack of cytokinin and auxin together limit organogenesis during the reproductive phase, supporting the proposal that the TMO5 acts downstream of MP to promote organogenesis via promoting both auxin and cytokinin signalling.

Given that TMO5 is itself auxin-regulated (De Rybel et al., 2013) and predominantly expressed sub-epidermally (this study), our data also suggest that in the inflorescence meristem at least, organogenesis involves an amplification process starting from an auxin trigger that encompasses multiple cell layers. This not only includes the concentration of auxin locally via PIN1-mediated transport (Bhatia et al., 2016), but also via the local induction of TMO5-mediated auxin synthesis, transport or signalling. As it has been established that the cytokinin synthesis genes *LONELY GUY3* (*LOG3*) and *LOG4* are targets of LHW, which heterodimerizes with TMO5 and its homologues in the shoot (Mor et al., 2022), our work also suggests LOG3 and LOG4 contribute to flower formation in addition to promoting meristem function (Chickarmane et al., 2012). Whether TMO5-induced cytokinin in turn promotes organ growth directly or indirectly by maintaining meristem cell divisions (Yoshida et al., 2011; Kong et al., 2024) is not yet clear, although evidence suggests a

direct role in this process (Besnard et al., 2014). Future work will aim to better understand how TMO5 and its homologues feed back to auxin signalling and how auxin and cytokinin ultimately work in conjunction to promote organ outgrowth, by identifying the genes involved.

## MATERIALS AND METHODS
### Plant materials and growth conditions
*Arabidopsis thaliana* ecotype Landsberg *erecta* (L*er*) or Columbia-0 (Col-0) was used as the wild type in this study. The *mp*-T370 mutant allele is in the L*er* ecotype and *mp*-B4149 allele is in the *Utrecht* ecotype (Bhatia et al., 2016; Hardtke and Berleth, 1998; Weijers et al., 2006), both *mp* alleles were used to exclude ecotype-specific effects The remaining mutants used in this study – including *tmo5*, *t5l1*, *t5l2* and *t5l3* (De Rybel et al., 2013) as well as *t5l4* (AT2G40200, SALK_008269.47.60, N508269) (this study) – are in Col-0 background. The *t5l4* mutant was genotyped using the primers listed in Table S1. The *tmo5,t5l1* double and *tmo5,t5l1,t5l2,t5l3* quadruple mutants have been described previously with genotyping details (De Rybel et al., 2013). The *tmo5,t5l2,t5l3* triple mutants were derived from *tmo5,t5l1,t5l2,t5l3* quadruple mutant siblings, where *t5l1* was segregating. The *tmo5,t5l4* double mutant was generated by crossing the respective single mutants and the *tmo5, t5l1,t5l4* triple mutant was produced by crossing *tmo5,t5l4* and *tmo5,t5l1* double mutants. All mutant combinations were genotyped using the primers listed in Table S1. The T5L3::n3GFP reporter has been described previously (De Rybel et al., 2013). Other transgenic lines include *mp*-T370 mutants carrying the pPIN1::PIN1-CFP reporter, which were transformed with either TMO5::TMO5-2YPET, TMO6::TMO6-2YPET, DOF5.8::DOF5.8-YPET or a genetic mosaic construct incorporating TMO5, TMO6 and DOF5.8 (details provided below). Additionally, the translational reporters T5L1::T5L1-2YPET T5L2::T5L2-2YPET or T5L4::T5L4-2YPET were introduced into wild-type plants carrying pPIN1::PIN1-GFP. Newly generated plant lines are available upon request.

Seeds were surface sterilized with 70% (v/v) ethanol for 10 min. The seeds were then put on a sterile filter paper to dry out and to remove ethanol content. The sterilized seeds were plated on 1× Murashige and Skoog (MS) basal salt mixture (Sigma-Aldrich, M5524), 1% sucrose, 0.5 g/L MES 2-(MN-morpholino)-ethane sulfonic acid (Sigma-Aldrich, M2933), 0.8% Bacto Agar (BD Biosciences) and 1% MS vitamins (Sigma-Aldrich, M3900). The pH was adjusted to 5.7 with 1 M KOH. After 2 days of stratification at 4°C, the seed plates were moved to the growth room at 22°C under continuous light. For imaging of wild-type inflorescence meristems, plants were grown on soil at 18°C in short-day conditions (16 h dark/8 h light).

### Construction of reporters and transgenic plants
For genetic mosaic analyses using the CRE/Lox system, we generated stable transgenic lines containing a template for sectoring *TMO5* (UBQ10p:lox spacer lox:GENE-2YPET) or *DOF5.8* or *TMO6* (UBQ10p:lox spacer lox: GENE-YPET) along with a dexamethasone-inducible CRE Recombinase (CLV3p:CRE-GR) (Bhatia et al., 2016). Here, 'GENE' refers to *DOF5.8*, *TMO5* or *TMO6*. To create UBQ10p:lox spacer lox:GENE-2xYPET, we initially cloned a 4.6 kb SfiI-BamHI fragment from UBQ10p:lox spacer lox: MP-VENUS (Bhatia et al., 2016). This fragment was inserted upstream of a 9× alanine linker followed by single YPET (Nguyen and Daugherty, 2005) or two tandem copies of YPET and the OCS terminator. *DOF5.8* and *TMO6* or *TMO5* genomic coding sequences were then amplified using specific primers (Table S1) and cloned into a BamHI site, fused in-frame with the YPET tag to generate UBQ10p:lox spacer lox:GENE-YPET or UBQ10p:lox spacer lox:GENE-2xYPET. Finally, UBQ10p:lox spacer lox:GENE-2xYPET and CLV3p:CRE-GR spacer lox were combined into the transfer DNA vector BGW (Karimi et al., 2002) using Gateway technology (Invitrogen). These constructs were then transformed into *Agrobacterium* strain C58C1 via electroporation and subsequently introduced into *Arabidopsis* plants using the floral dipping method (Clough and Bent, 1998).

Genomic sequences of *DOF5.8*, *TMO6*, *TMO5*, *T5L1*, *T5L2* or *T5L4* containing upstream and coding sequences were then amplified using specific primers (Table S1) and cloned into XhoI and BamHI sites, fused

in-frame with a 9× alanine linker followed by YPET or 2xYPET tag to generate DOF5.8-YPET, TMO6-2YPET, TMO5-2YPET, T5L1::T5L1-2YPET, T5L2::T5L2-2YPET or T5L4::T5L4-2YPET translational reporter genes. Translational reporter genes were transferred as NotI restriction fragments to T-DNA vector pMLBART (Gleave, 1992). Newly generated plasmids will be made available upon request.

### Chemical treatments
To induce *DOF5.8/TMO5/TMO6* sectors in the *mp* mutant meristem, *mp* mutants harbouring the respective sectoring constructs (CLV3p::CRE-GR+UBQ10p::lox spacer lox::GENE-2YPET) were germinated and grown on 1× MS agar medium until the emergence of leafless or flowerless dome structures. A 10-20 µl aliquot of 10 µM dexamethasone (DEX) in sterile water (prepared from a 10 mM stock dissolved in absolute ethanol) was directly applied to the dome meristem. Imaging was performed using confocal or brightfield microscopy at different time intervals, starting from 0 day and continuing up to 6-14 days. For the induction of the same constructs in the wild-type vegetative meristems, seeds were germinated on MS medium containing 10 µM DEX. The vegetative meristem was dissected 3 days after stratification (das), following the method described previously (Caggiano et al., 2021). Seedlings displaying sector formation were transferred to MS medium without DEX, and the meristem was imaged until 5-6 das. For overexpression of TMO5 (UBQ10p::lox spacer lox::TMO5-2YPET) in *mp* mutants, seedlings having dome meristems grown on MS medium were transferred to medium containing 10 µM DEX for continuous induction. Confocal or brightfield imaging was performed over a 6-14 day period. For overexpression in wild-type vegetative meristem, seedlings were germinated and grown on MS medium containing 10 µM DEX. Vegetative meristems were dissected at 3 das and imaged until 5 das. In all experiments, *mp* mutants were imaged before and after induction.

For NPA treatment on the *mp* mutant (to inhibit polar auxin transport), the dome meristem harbouring the *TMO5* sectoring construct (CLV3p::CRE-GR+UBQ10p::lox spacer lox::TMO5-2YPET) was pre-treated with 10 µM NPA for 2 days before DEX induction to eliminate residual auxin transport. After a single 10 µM DEX treatment, the mutants were continuously grown in the presence of 10 µM NPA or a DMSO mock treatment. Imaging was performed before and after chemical treatments and continued until 10 days post DEX treatment.

For NPA treatment in wild type and *tmo5/t5l* single and higher-order mutants, seedlings were grown from germination on MS medium containing 1 µM NPA or a DMSO mock treatment. Vegetative and inflorescence meristems were imaged using brightfield microscopy.

For auxinole treatment of the *mp* mutant (to inhibit auxin signalling), the dome meristem harbouring the *TMO5* sectoring construct was pre-treated with 100 µM auxinole or a DMSO mock for 2 days to eliminate residual auxin signalling. Following a single 10 µM DEX treatment on the dome meristem, the mutants were continuously grown in either 100 µM auxinole or a mock treatment. Imaging was performed before and after chemical treatments and continued until 10 days post-DEX treatment.

For organ outgrowth in the *mp* mutant, cytokinin (10 µM TZ) was applied to the *mp* vegetative meristem, while both auxin (5 mM IAA) and cytokinin (1 mM TZ) were applied to young pin-like inflorescence meristems, with DMSO/water as the mock control. Observations were recorded 7-14 days after treatment.

For auxin and cytokinin treatment on *tmo5* quadruple mutants (*tmo5,t5l1,t5l2,t5l3*), a single application of 10 µM TZ and 1 µM NAA was administered directly to the shoot tip of 5-day-old seedlings. Treatments were applied either individually or in combination, with DMSO/water as the mock control. Additionally, 1 µM NAA or 1 µM or 0.5 µM TZ was included in the growth medium under some conditions. Observations were taken 10-20 days after treatment.

### Inflorescence floral quantitation
To quantify the number of flowers in the inflorescence meristem in wild type and *tmo5* mutants, primary inflorescences were examined at the stage when the first flower bud had opened. The total number of flower buds within the inflorescence meristem was counted, and the average number of flower buds per inflorescence was calculated.

## Phyllotaxy measurement

To measure the phyllotactic pattern in wild type and *tmo5* mutants, the divergent angles between successive siliques in the mature inflorescence were assessed. The angles between each pair of siliques were recorded and averaged for comparison.

## Confocal live imaging

Wild-type vegetative and inflorescence meristems were dissected for imaging as previously described (Caggiano et al., 2021; Heisler et al., 2005). Confocal live imaging was performed using a Leica TCS-SP5 upright laser scanning confocal microscope with hybrid detectors (HyDs). Imaging used a 25× water objective (N.A. 0.95) or a 63× objective (N.A. 1.20) and a pixel format of either 512×512 or 1024×1024 for enhanced resolution. The system was configured for bidirectional scanning at speeds of 400 Hz or 200 Hz, with line averaging set to 2 or 3, producing optical sections with a thickness of 1 μm.

The laser settings for GFP and YPet/VENUS, as described in previous studies (Kareem et al., 2022), were applied. CFP was excited using 456 nm laser and emission window was set to 465-500 nm using argon laser. The pinhole was adjusted based on fluorescence brightness to avoid signal saturation or bleaching. A smart gain setting of 100% was employed, and sequential scan mode was used to switch between frames for imaging CFP/ GFP and YPet simultaneously.

## Image analysis and data processing

Images were analyzed using Imaris 9.1.2 (Bitplane) or ImageJ (FIJI, https:// fiji.sc). They were annotated and arranged in Adobe Photoshop (2020) and Affinity Designer 1.10.8. PIN1 polarity assessments were performed by examining arcs of fluorescent signal that extend beyond cell junctions and around cell corners, as previously described (Kareem et al., 2022). Statistical tests used are included in the figure legends.

## Acknowledgements
We thank Dolf Weijers and Bert De Rybel for providing seed for the *t5l4* mutant as well as published DNA and plant lines.

## Competing interests
The authors declare no competing or financial interests.

## Author contributions
Conceptualization: M.G.H., A.K.; Funding acquisition: M.G.H.; Investigation: M.G.H., A.K., C.O.; Methodology: A.K.; Project administration: M.G.H.; Resources: M.G.H., C.O.; Supervision: M.G.H., C.O.; Writing – original draft: M.G.H., A.K.; Writing – review & editing: M.G.H., A.K.

## Funding
This work was supported by the Australian Research Council Discovery Grant DP180101149 awarded to M.G.H. Open Access funding provided by University of Sydney. Deposited in PMC for immediate release.

## Data and resource availability
All relevant data and details of resources can be found within the article and its supplementary information.

## The people behind the papers
This article has an associated 'The people behind the papers' interview with some of the authors.

## Peer review history
The peer review history is available online at https://journals.biologists.com/dev/lookup/doi/10.1242/dev.205255.reviewer-comments.pdf

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
