## [Peer Review File · Development (Cambridge, England)]

TM05 regulates PIN1 polarity convergence and organogenesis downstream of MONOPTEROS in the *Arabidopsis* shoot

Abdul Kareem, Carolyn Ohno and Marcus G. Heisler

DOI: 10.1242/dev.205255

Editor: Dominique Bergmann

Review timeline

Original submission: 29 November 2024

Editorial decision: 13 January 2025

Resubmission: 19 September 2025

Editorial decision: 27 October 2025

First revision: 5 November 2025

Accept: 7 November 2025

Original submission

First decision letter

MS ID#: dev.204569

MS TITLE: Vascular Transcription Factor TMO5 Regulates PIN1 Polarity and Organogenesis Downstream of MONOPTEROS in the Shoot Meristem

AUTHORS: Marcus G Heisler; Abdul Kareem; Carolyn Ohno

Dear Dr Heisler,

I have now received all the referees reports on the above manuscript, and have reached a decision. The referees' comments are appended below, or you can access them online: please go to:

As you will see from their reports, the referees recognise the potential of your work, but they also raise significant concerns about it. Given the nature of these concerns, I am afraid I have little choice other than to reject the paper at this stage.

However, having evaluated the paper, I do recognise the potential importance of this work. I would therefore be prepared to consider as a new submission an extension of this study that contains new experiments, data and discussions and that address fully the major concerns of the referees. The work required goes beyond a standard revision of the paper. Please bear in mind that the referees (who may be different from the present reviewers, although I will try to engage the same reviewers) will assess the novelty of your work in the context of all previous publications, including those published between now and the time of resubmission.

Comments from the Reviewers:

Reviewer 1: SUMMARY OF THE ADVANCE MADE IN THIS PAPER AND ITS POTENTIAL SIGNIFICANCE TO THE FIELD

In this manuscript, the authors investigate how the activity of MP and TMO5 induces changes in PIN1 polarity and organogenesis. The novel aspect of this work lies in enhancing understanding of how does transcriptional regulation (MP and TMO5) induce changes in protein localization (PIN1) non cell

autonomously. The manuscript is well written and interesting. Some remarks and suggestions are listed below.

SUGGESTIONS TO AUTHORS

Line 82: In fig1 and following figures: the color scale only represents the CFP expression, not that of the YPET. Please add or explain in the legend.

Line 76. - Was there any other reason that the authors chose to focus on TMO5, TMO6 and DOF5.8 or rather purely based on previous work where it was found as a downstream target.

Line 80-84 and Fig 1 - Fig S1 - quantifying "partial overlap" - is there a way to show quantifications for these? It is also not clear from the Figure panels that there is a difference in expression of TMO5 and DOF5.8 vs. TMO6. Can the authors make this more clear? Is TMO5 overlapping more than DOF5.8 with PIN1? In 1B it is not clear that DOF5.8 overlaps with PIN1, except maybe for the bit of plant seen in the left corner. Is it possible to get a better resolution image (more magnification) to show the overlap or signal intensity quantification showing the overlap? Related, the cross sections are not drawn in the same way for Panels A-B-C of Figure1. In panel A the cross section goes from primordia to primordia; while in B it goes from primordia to in-between two primordia.

Line 92 - "reduced compared to WT" - is it possible to quantify difference of TMO6 reporter signal in the two backgrounds to support this claim.

Line 92-94 - Expression correlates with MP activity - it seems that it correlates with proper organogenesis, as there is still expression in the leaves of the mp mutant and that MP activity is necessary for proper organogenesis...consider rephrasing? Also, there is no expression in the zones that will become the primordia, so the expression is following the organogenesis program rather than defining the future site. Does this affect the author's conclusions?

Line 105 - Figure S2, panel H - if possible, to replace this image with a 5 day old plant, to provide better comparison to panel F and C, however if this is difficult then no problem.

Line 105: why were TMO5 clones induced in the mpB4149 mutant and DOF and TMO5 clones in the mpT370 mutant? Is the correlation between the choice of mutant and the success of organogenesis accidentally or does the genotype matter? If so, the experiments would need to be repeated in the same mutant background to allow correct comparison.

Line 112 - subepidermal cells and PIN1 expression. The authors claim that TMO5 induces a localized PIN1 polarity convergence in a non-cell autonomous manner, as seen in Fig2O and P and S2J, with PIN1 polarizing towards TMO5 in adjacent cells. However, when looking at PIN1 expression at the subepidermal later (2Q and S2K) one does not see PIN1 get polarized towards the subepidermal cells expressing TMO5. Rather PIN1 is getting polarized towards the new organ (flower) outgrowth, as is expected for this auxin transporter. The claim that TMO5 is necessary to promote organ outgrowth downstream of MP is well supported, however the claim that TMO5 polarizes PIN1 non cell autonomously is not supported by Fig1 A, FigS1 A, FigS2 K and Fig2 Q. In order to make this claim, the authors would have to provide further evidence, for example by co-expression assays. Perhaps the authors refer to PIN1 localization converging only in epidermal cells toward the outgrowth, please clarify/rephrase.

Line 123 - The authors refer to Fig S3A to show they found no defects of organ formation for tmo5 single. But there is no image of tmo5 in control conditions in panel A, only a T-DNA insertion for T5L4.

Line 127 - Newly identified homologue T5L4 - would it be possible to go into more details on gene similarity or a evolutionary tree to see relation/level of similarity to the other TMO5 family members

Line 131: T5L4 expression seems almost mutually exclusive from PIN1 in Fig 3D and different from the other paralogs. Should this be included in the text?

Line 136 and Fig 3E - As this is count data, a students t-test is not the best statistical tool to use. Consider using Poisson Regression. In addition, it would be better to use a box and violin plot and not a bar chart to show these results. The same for S3 B.

Fig 3H - Could the axes of this graph be labelled more clearly? The legend says that WT has a divergent angle of 137, however there are blue bars also at 90 and 175.

Line 148 - Below, the authors explain that auxinole is an inhibitor of auxin signaling (line 151). Could the authors explain for Line 148 in a similar manner what does NPA inhibit by inserting a phrase such as , "...presence of 10µM NPA, an auxin transport inhibitor (Abas 2020 or another reference), a concentration of NPA sufficient to cause.." to demonstrate clearly the difference and logic of using 2 different auxin inhibitor types.

Line 153 - The authors can consider adding "...requires residual auxin signaling in the mp mutant to induce organ formation, albeit being independent of auxin transport..." or something among those lines to highlight why there is a difference when using different auxin inhibitors.

Line 154 - rephrase and specify. Suggestion: To test whether TMO5 function is also sensitive to changes in auxin transport properties.

Line 155 - combinations with upon 1 μ M NPA treatment - rephrase or delete the words upon and treatment

Fig 3 I-L - Back to the PIN1 polarization being based non cell autonomously on TMO5, after treatment with NPA and auxinole, does one see an effect on polarity in presence of TMO5?

line 164 - change division to divisions

Fig 4 - legend title "Cytokinin and Auxin Substitute TMO5/T5L Function for Organ Formation". The word "Substitute" suggests that exogenous hormone application can fully rescue the phenotype caused by a lack of functional TMO5/T5L in the mp and higher order tmo5 mutants. Either the authors have to provide an additional control (wild type plant like Col-0) to demonstrate that the number of leaves and flowers per plant and the outgrowth in auxin and cytokinin treated mutants plants is the same as wt or rephrase this, for example to say 'partial rescue'.

Fig4 C,E,F - would be better to show a violin plot for quantitative data then a bar graph. In addition for quantitative data students t-test is not the most fitting choice.

Line 170;line 181 and Fig 4D-J. - The authors use both exogenous IAA (fig 4D) and NAA (fig 4E,F) - what is the reason for the different types? NAA is a synthetic auxin and thereby produces more outgrowth as is more stable than IAA. Could the authors mention that they use different types in the text and expand on this rather than just referring to 'auxin' and 'cytokinin'? In addition Fig 4 just refers to 'auxin and cytokinin' however details such as what exogenous treatment and what molarity was applied are missing from the graphs. If one uses a higher molarity most likely growth would still be arrested - perhaps the authors can expand on why they chose specifically those concentrations.

Line 171,174 - the word rescue suggests that the plant can grow normally and reach maturity as a wild type plant. Can the hormone application fully rescue the organ growth defect or does it still arrest growth and remain dwarfed? Rescue suggests that the plant could even produce viable seeds? Please rephrase.

Line 163 - 184. - This paragraph and the accompanying Fig 4 demonstrate that mp and tmo5,t5l1,t5l2,t5l3 mutants are affected by application of 5mM IAA, 1 μ M NAA and 10 μ M trans-zeatin in their organ formation properties, however for the authors to demonstrate full rescue of the vegetative growth and substitution of TMO5/T5L they would have to show an older plant (for example 2 month old) and a wild type plant. The authors do not indicate the age of the plants in Figure 4 or show a wild type control. Please rephrase, as exogenous hormone application usually has a very unspecific effect on plant development and can either promote growth or delay growth depending on the concentration and thus does not add much to support the main findings of this paper, which show SAM TMO5 having an effect on PIN1 polarity and organogenesis.

The authors refer to the tmo5 quad mutant in the figure legends and methods, could it also once be mentioned in the main text to clarify exactly what mutant that is.

Line 186-203: Although I agree that auxin and cytokinin are required for flower formation and these are intimately linked to TMO5 function, the authors should also consider that cytokinin is the main downstream effect of TMO5 function. Cytokinin has also been shown to affect auxin biosynthesis, signaling and transport. Thus, disrupting TMO5 levels will affect cytokinin levels and this mobile signal will affect auxin in many ways. Thus, the effects seen on PIN polarity could be indirectly caused by changing cytokinin levels. Does this affect the authors conclusions? Can this thought be incorporated in the discussion?

Reviewer 2: SUMMARY OF THE ADVANCE MADE IN THIS PAPER AND ITS POTENTIAL SIGNIFICANCE TO THE FIELD

This manuscript provides valuable insights into the role of TMO5 in regulating PIN1 polarity and organogenesis in the shoot meristem. The authors demonstrate that TMO5 acts downstream of MONOPTEROS (MP) to coordinate hormonal signaling, specifically auxin and cytokinin, to drive organ initiation. Key findings include the non-cell-autonomous regulation of PIN1 polarity by TMO5 and its ability to promote organ outgrowth in both vegetative and inflorescence meristems. Additionally, the study highlights the synergistic roles of TMO5 and its homologs (T5L1-T5L4) in ensuring proper organ positioning and development, making a significant contribution to the field of plant developmental biology.

SUGGESTIONS TO AUTHORS

Quantitative Analysis of Spatial Gene Expression Patterns

In Fig. 1, the arguments regarding the spatial patterns of gene expression rely solely on representative images and qualitative observations, which may be insufficient to fully support the conclusions. To enhance the robustness of the findings, it would be helpful to include quantitative or statistical analyses, such as the overlap ratio with PIN1 expression patterns. Similarly, for the reduced expression levels in the mp mutant background, quantitative measurements would strengthen the interpretation rather than relying solely on visual observations.

Detailed Analysis of TMO5 Homologs

From Fig. 3A-D, it seems that the expression patterns of T5L1, T5L2, T5L3, and T5L4 in the shoot meristem might differ. While genetic analyses suggest that these homologs may have redundant functions, it would be helpful to clarify whether they exhibit distinct spatiotemporal expression patterns at the cellular level. Alternatively, do these homologs show co-expression at the cellular level? Addressing these questions would provide a deeper understanding of their roles and how they contribute to organogenesis.

Minor Points

The references are inconsistently formatted, which detracts from the overall presentation. Please ensure that the reference formatting is consistent throughout the manuscript.

The paragraph beginning at line 186 contains mixed fonts. We recommend thoroughly checking the manuscript for any formatting issues before resubmission.

Reviewer 3: SUMMARY OF THE ADVANCE MADE IN THIS PAPER AND ITS POTENTIAL SIGNIFICANCE TO THE FIELD

This manuscript by Kareem and colleagues uses a combination of molecular genetics, including a clonal analysis, and live imaging to analyze the contribution of genes regulated downstream of auxin signaling to the polarities of PIN1 in the shoot apical meristem. Some of the authors have described previously a contribution of the MP signaling effector to regulating PIN1 polarities and they explore here the possibility that a direct target of MP, TMO5, mediates at least some of the effect of MP on PIN1 polarities. The authors address a question essential to understand plant development. However the results do not fully support the conclusions of the authors. One of the main issue is that it takes several days to form organs after induction of sectors with TMO5, suggesting that TMO5 probably only plays a very indirect role in PIN1 polarity. It seems exaggerated to conclude that TMO5 'regulate' PIN1 polarity and it is unclear what we really learn from this work.

SUGGESTIONS TO AUTHORS

Main concerns:

- 1- A general and important concern the data and their presentation. Most of the conclusions are derived from single images (from microscopy or showing phenotypes) in the figures. There is no information about the N for experiments reported in Fig 1, S1, 2, 3 (microscopy images and F,G), S3 (images in C-L), S4. The authors need to provide this information and extra images (see below). Without this, their claim are not supported by the data.
- 2- Lines 75-94 - Previous publications including one from some of the authors have analyzed the expressions of TMO5 and DOF5.8, and the dependency of their expression on MP. They should be discussed here: Ram et al PLoS genetics 2020, Larrieu et al iScience 2022, Mor et al iScience 2022.
- 3- Lines 96-121 and corresponding figures - In line with comment #1, with single clones being shown on Figures, it is not possible to evaluate the solidity of the results. N and statistics are needed here for all the observations.
- 4- Lines 111 and 116 - The authors mention PIN1 polarity convergence induced by the presence of a TMO5 clone. The PIN1 signal cannot be easily attributed to a specific cells in such images. Can't the authors quantify the polarity to backup their claim? And on Fig 2M,N what is the most obvious is a correlation between high PIN1 and the TMO5 clone. What is the timing of the PIN1 accumulation and possible effect on polarity in relation to the appearance of TMO5? And how does this relate to the organogenesis in term of timing?

5- As suggested by Fig 2 and S2, the growth induced by TMO5 clones and effects on PIN1 would take several days. This suggests a largely indirect feedback effect of TMO5 on organogenesis/PIN1 polarities. These experiments do not support the idea of a role 'for TMO5 in controlling PIN1 polarity'. They only suggest that TMO5 can feedback indirectly onto organogenesis/PIN1 polarities. The claim from the author that they have identified a role for TMO5 in the regulation of PIN1 polarity is not supported by the result of their analysis.

6- From line 123-142 - Mor et al iScience 2022 report some shoot meristem defects in TMO5/TMO5-like mutants. This should be discussed.

7- Lines 163-184: It is unclear why the authors talk about 'organ outgrowth defects' in the tmo5/t5l quadruple mutants. The mutant can clearly make leaves. Isn't it simply that it grows more slowly? The results of the authors suggest that CK alone is sufficient to accelerate its growth and auxin appears to have a synergistic effect on growth. This would explain why more flowers are observed with auxin+CK on Figure 4. In any case the simple fact that the quadruple mutant can already make leaves does not allow to conclude that 'cytokinin can substitute for the role of TMO5/T5L ...' (lines 182-184).

Other concerns:

8- Fig3A: the SAM seems abnormally small.

9- Fig 4A-C - a representative situation for the mp would a mutant with 4 leaves (Fig 4C). The authors should put a representative image in Fig 4A.

Response to Reviewers

Reviewer 1: SUMMARY OF THE ADVANCE MADE IN THIS PAPER AND ITS POTENTIAL SIGNIFICANCE TO THE FIELD

In this manuscript, the authors investigate how the activity of MP and TMO5 induces changes in PIN1 polarity and organogenesis. The novel aspect of this work lies in enhancing understanding of how does transcriptional regulation (MP and TMO5) induce changes in protein localization (PIN1) non cell autonomously. The manuscript is well written and interesting. Some remarks and suggestions are listed below.

SUGGESTIONS TO AUTHORS

1.1 Line 82: In fig1 and following figures: the color scale only represents the CFP expression, not that of the YPET. Please add or explain in the legend.

The legends have been updated.

1.2. Line 76. - Was there any other reason that the authors chose to focus on TMO5, TMO6 and DOF5.8 or rather purely based on previous work where it was found as a downstream target.

This study represents an initial effort to identify targets of MP relevant to organogenesis and PIN1 polarity regulation. Given many MP targets have already been identified, we decided to investigate these genes as a starting point. Our introductory text now explains this more clearly.

1.3. Line 80-84 and Fig 1 - Fig S1 - quantifying "partial overlap" - is there a way to show quantifications for these? PIN1 is expressed very broadly, in fact the majority of signal is in the epidermis in non-primordium regions. PIN1 is also membrane localized while the other factors we focus on are nuclear localized. Hence we do not quantify the degree of signal overlap because this measurement would not be relevant to whether there is a coincidence of signal at primordium positions. However, from Fig. 1 and S1, the coincidence and relative timing between TMO5 and high levels of PIN1 (which mark primordium positions) can be clearly seen qualitatively and we think this is sufficient to support our conclusions.

1.4 It is also no clear from the Figure panels that there is a difference in expression of TMO5 and DOF5.8 vs. TMO6. Can the authors make this more clear?

New panels of individual channels for Fig1A-C longitudinal sections are now included with arrowheads to point out the main differences.

1.5 Is TMO5 overlapping more than DOF5.8 with PIN1?

They both overlap similarly although DOF5.8 is somewhat broader. This difference is now mentioned in the text.

1.6 In 1B it is not clear that DOF5.8 overlaps with PIN1, except maybe for the bit of plant seen in the left corner. Is it possible to get a better resolution image (more magnification) to show the overlap or signal intensity quantification showing the overlap? Related, the cross sections are not drawn in the same way for Panels A-B-C of Figure1. In panel A the cross section goes from primordia to primordia; while in B it goes from primordia to in- between two primordia.

A new cross section, drawn from primordium to primordium as in A and C, is now included and shows more clearly the overlap between DOF5.8 and PIN1.

1.7 Line 92 - "reduced compared to WT" - is it possible to quantify difference of TMO6 reporter signal in the two backgrounds to support this claim.

This sentence has been rephrased to more accurately reflect the difference expression domain rather than expression level.

1.8 Line 92-94 - Expression correlates with MP activity - it seems that it correlates with proper organogenesis, as there is still expression in the leaves of the mp mutant and that MP activity is necessary for proper organogenesis...consider rephrasing?

The sentence has been rephrased.

1.9 Also, there is no expression in the zones that will become the primordia, so the expression is following the organogenesis program rather than defining the future site. Does this affect the author's conclusions?

While it is true that high PIN1 levels mark primordium sites 1 plastochron earlier than TMO5 (I2), this high PIN1 expression is localized to the epidermis and the convergence pattern of polarity at this stage, is still being formed (Heister et al., 2005). Local high expression of PIN1 in subepidermal cells only occurs a plastochron later, at I1 - coinciding with the onset of TMO5. So, while it is true that TMO5 is likely not involved in initial steps of specification, the evidence indicates it coincides with the onset of PIN1, to promote organ outgrowth, in sub- epidermal regions. According to the current literature, organ growth would be expected to help orient PIN1 via mechanical stresses and so a dual role in promoting organ growth and reinforcing auxin localization via changes to PIN1 polarity would be expected. This is in line with our conclusions.

1.10 Line 105 - Figure S2, panel H - if possible, to replace this image with a 5 day old plant, to provide better comparison to panel F and C, however if this is difficult then no problem.

Unfortunately this is not possible.

1.11 Line 105: why were TMO5 clones induced in the mpB4149 mutant and DOF and TMO5 clones in

the mpT370 mutant? Is the correlation between the choice of mutant and the success of organogenesis accidentally or does the genotype matter? If so, the experiments would need to be repeated in the same mutant background to allow correct comparison.

The data indicates there is no ecotype specificity since we tested TMO5 in both mpB4149 as well as in mpT370 and observed the same response (e.g. compare Fig. 2 A-C vs Fig. S2 A-D), although we did not test DOF or TMO6 in mpB4149.

1.12 Line 112 - subepidermal cells and PIN1 expression. The authors claim that TMO5 induces a localized PIN1 polarity convergence in a non-cell autonomous manner, as seen in Fig2O and P and S2J, with PIN1 polarizing towards TMO5 in adjacent cells. However, when looking at PIN1 expression at the subepidermal later (2Q and S2K) one does not see PIN1 get polarized towards the subepidermal cells expressing TMO5. Rather PIN1 is getting polarized towards the new organ (flower) outgrowth, as is expected for this auxin transporter. The claim that TMO5 is necessary to promote organ outgrowth downstream of MP is well supported, however the claim that TMO5 polarizes PIN1 non cell autonomously is not supported by Fig1 A, FigS1 A, FigS2 K and Fig2 Q. In order to make this claim, the authors would have to provide further evidence, for example by co-expression assays. Perhaps the authors refer to PIN1 localization converging only in epidermal cells toward the outgrowth, please clarify/rephrase.

The polarity of PIN1 in subepidermal cells is very difficult to determine due to low expression levels and therefore we do not claim the subepidermal cells surrounding subepidermal TMO5 expressing cells in Fig. 2Q show polarity in any particular direction. We now make this clear in the text describing these data.

In our 2016 paper examining the influence of MP on PIN1 polarity, the earliest we see changes in PIN1 polarity in response to MP clones is around 1-2 days. This timeframe is consistent with our earlier proposal that the influence of auxin on PIN1 and microtubule polarities/orientations occurs via changes in cell wall properties and mechanical stress, since such changes would take time given that MP is a transcription factor. Given TMO5 is also a transcription factor, we would expect a similar indirect regulatory relationship to PIN1 polarity changes. In fact, for TMO5 we find a similar delay between clone appearance and polarity change compared to MP, i.e. between 1-2 days. Is the influence of TMO5 on PIN1 polarity downstream of primordium specification? We now include new data that demonstrate TMO5 is able to cause changes to PIN1 polarity that are not linked to primordium development. For instance, clones that never develop into primordia (Fig. 2N) or clones consisting of only two cells where PIN1 signal is primarily localized to the membranes dividing the two adjacent cells only (Fig 3T observed in the presence of NPA). These new examples are referenced in the text to help clarify (lines 122 and 175).

1.13 Line 123 - The authors refer to Fig S3A to show they found no defects of organ formation for tmo5 single. But there is no image of tmo5 in control conditions in panel A, only a T-DNA insertion for T5L4.

Thanks, this was a typo. It has now been corrected to Fig.S3B.

1.14 Line 127 - Newly identified homologue T5L4 - would it be possible to go into more details on gene similarity or a evolutionary tree to see relation/level of similarity to the other TMO5 family members

More information on T5L4 can be found in De Rybel et al., 2013 (also referenced in our text). We feel that delving into a phylogeny is outside the scope of this very brief format paper.

1.15 Line 131: T5L4 expression seems almost mutually exclusive from PIN1 in Fig 3D and different from the other paralogs. Should this be included in the text?

This has now been included in the text

1.16 Line 136 and Fig 3E - As this is count data, a student's t-test is not the best statistical tool to use. Consider using Poisson Regression. In addition, it would be better to use a box and violin plot and not a bar chart to show these results. The same for S3 B.

We replaced the bar graph with a box plot to better show data distribution in Fig. 3E and S3B. Also performed Poisson regression as suggested.

1.17 Fig 3H - Could the axes of this graph be labelled more clearly? The legend says that WT has a divergent angle of 137, however there are blue bars also at 90 and 175.

We have modified the legend. For the X axis, rather than "Divergent angle" we have written "Divergence angle". The Y axis is now labelled "Frequency of angle (%)" and the legend has been modified to ascribe a 137 degree divergence angle to the majority of plants rather than being absolute.

1.18 Line 148 - Below, the authors explain that auxinole is an inhibitor of auxin signaling (line 151). Could the authors explain for Line 148 in a similar manner what does NPA inhibit by inserting a phrase such as, "...presence of 10 μ M NPA, an auxin transport inhibitor (Abas 2020 or another reference), a concentration of NPA sufficient to cause..." to demonstrate clearly the difference and logic of using 2 different auxin inhibitor types.

This has been added to the text

1.19 Line 153 - The authors can consider adding "...requires residual auxin signaling in the mp mutant to induce organ formation, albeit being independent of auxin transport..." or something among those lines to highlight why there is a difference when using different auxin inhibitors.

We have added in brackets "(not transport)" to more clearly contrast the two experiments, although we also feel this clarification should not really be needed.

1.20 Line 154 - rephrase and specify. Suggestion: To test whether TMO5 function is also sensitive to changes in auxin transport properties.

We have rephrased the sentence

1.21 Line 155 - combinations with upon 1 μ M NPA treatment - rephrase or delete the words upon and treatment

(updated to line 162) This sentence has been edited

1.22 Fig 3 I-L - Back to the PIN1 polarization being based non cell autonomously on TMO5, after treatment with NPA and auxinole, does one see an effect on polarity in presence of TMO5?

We did not see any polarization after auxinole treatment but we did after NPA treatment. However this polarization was only associated with small clones suggesting that NPA reduces the degree of polarization in some way. These findings are now included in the text (lines 364-365).

1.23 line 164 - change division to divisions

This has been corrected.

1.24 Fig 4 - legend title "Cytokinin and Auxin Substitute TMO5/T5L Function for Organ Formation".

The word "Substitute" suggests that exogenous hormone application can fully rescue the phenotype caused by a lack of functional TMO5/T5L in the mp and higher order tmo5 mutants. Either the authors have to provide an additional control (wild type plant like Col-0) to demonstrate that the number of leaves and flowers per plant and the outgrowth in auxin and cytokinin treated mutants plants is the same as wt or rephrase this, for example to say 'partial rescue'.

We will have rephrased the legend title.

1.25 Fig4 C,E,F - would be better to show a violin plot for quantitative data than a bar graph. In addition for quantitative data students t-test is not the most fitting choice.

We replaced the bar graph with a box plot to better show the data distribution in Fig. 4C, E, F. and have used the correct statistical test (Poisson regression for the count data and t-test for the rest).

1.26 Line 170;line 181 and Fig 4D-J. - The authors use both exogenous IAA (fig 4D) and NAA (fig 4E,F) - what is the reason for the different types? NAA is a synthetic auxin and thereby produces more outgrowth as is more stable than IAA. Could the authors mention that they use different types in the text and expand on this rather than just referring to 'auxin' and 'cytokinin'? In addition Fig 4 just refers to 'auxin and cytokinin' however details such as what exogenous treatment and what molarity was applied are missing from the graphs. If one uses a higher molarity most likely growth would still be arrested - perhaps the authors can expand on why they chose specifically those concentrations.

We provided the details of auxin and cytokinin treatments in both the text and the figure, in addition to the legend.

1.27 Line 171,174 - the word rescue suggests that the plant can grow normally and reach maturity as a wild type plant. Can the hormone application fully rescue the organ growth defect or does it still arrest growth and remain dwarfed? Rescue suggests that the plant could even produce viable seeds? Please rephrase.

The sentences have been rephrased

1.28 Line 163 - 184. - This paragraph and the accompanying Fig 4 demonstrate that mp and tmo5,t5l1,t5l2,t5l3 mutants are affected by application of 5mM IAA, 1 uM NAA and 10 uM trans-zeatin in their organ formation properties, however for the authors to demonstrate full rescue of the vegetative growth and substitution of TMO5/T5L they would have to show an older plant (for example 2 month old) and a wild type plant. The authors do not indicate the age of the plants in Figure 4 or show a wild type control. Please rephrase, as exogenous hormone application usually has a very unspecific effect on plant development and can either promote growth or delay growth depending on the concentration and thus does not add much to support the main findings of this paper, which show SAM TMO5 having an effect on PIN1 polarity and organogenesis. The authors refer to the tmo5 quad mutant in the figure legends and methods, could it also once be mentioned in the main text to clarify exactly what mutant that is.

We have refrained from using the term "rescue" and now refer more specifically to organ formation and growth. In regard to the quadruple mutant, we now mention the full name explicitly in the main text, e.g lines 390, 395, 457, 501, 503, 706, 709, 711.

1.29 Line 186-203: Although I agree that auxin and cytokinin are required for flower formation and these are intimately linked to TMO5 function, the authors should also consider that cytokinin is the main downstream effect of TMO5 function. Cytokinin has also been shown to affect auxin biosynthesis, signaling and transport. Thus, disrupting TMO5 levels will affect cytokinin levels and this mobile signal will affect auxin in many ways. Thus, the effects seen on PIN polarity could be indirectly caused by changing cytokinin levels. Does this affect the authors conclusions? Can this

thought be incorporated in the discussion?

Although alterations in auxin abundance may occur via TMO5-induced changes in cytokinin, we see a restoration of flower production in the quad mutant only when both cytokinin and auxin are applied, not cytokinin alone. Hence, this is evidence that TMO5 likely acts via auxin in addition to cytokinin, at least in the inflorescence meristem to help promote organogenesis.

Reviewer 2: SUMMARY OF THE ADVANCE MADE IN THIS PAPER AND ITS POTENTIAL SIGNIFICANCE TO THE FIELD

This manuscript provides valuable insights into the role of TMO5 in regulating PIN1 polarity and organogenesis in the shoot meristem. The authors demonstrate that TMO5 acts downstream of MONOPTEROS (MP) to coordinate hormonal signaling, specifically auxin and cytokinin, to drive organ initiation. Key findings include the non-cell-autonomous regulation of PIN1 polarity by TMO5 and its ability to promote organ outgrowth in both vegetative and inflorescence meristems. Additionally, the study highlights the synergistic roles of TMO5 and its homologs (T5L1-T5L4) in ensuring proper organ positioning and development, making a significant contribution to the field of plant developmental biology.

SUGGESTIONS TO AUTHORS

2.1 Quantitative Analysis of Spatial Gene Expression Patterns

In Fig. 1, the arguments regarding the spatial patterns of gene expression rely solely on representative images and qualitative observations, which may be insufficient to fully support the conclusions. To enhance the robustness of the findings, it would be helpful to include quantitative or statistical analyses, such as the overlap ratio with PIN1 expression patterns. Similarly, for the reduced expression levels in the *mp* mutant background, quantitative measurements would strengthen the interpretation rather than relying solely on visual observations.

In Fig. 1. Regarding the qualitative differences observed in the expression patterns of the three reporters for TMO5, DOF5.8 and TMO6, no quantitative measure for these differences was provided because the exact level of difference has no bearing on our subsequent experiments or conclusions. We have however now provided the number of shoots imaged for checking the qualitative consistency of the patterns. In regard to the levels of these genes in the *monopteros* mutant, we also now provide the numbers of apices checked for consistency. Again, only qualitative differences are required for us to conclude that MP plays an important role in their regulation.

2.2 Detailed Analysis of TMO5 Homologs

From Fig. 3A-D, it seems that the expression patterns of T5L1, T5L2, T5L3, and T5L4 in the shoot meristem might differ. While genetic analyses suggest that these homologs may have redundant functions, it would be helpful to clarify whether they exhibit distinct spatiotemporal expression patterns at the cellular level. Alternatively, do these homologs show co-expression at the cellular level? Addressing these questions would provide a deeper understanding of their roles and how they contribute to organogenesis.

We feel that an in-depth investigation into any differences in function between these genes is appropriate for a new study, given the short length and scope of this Development Research Report,

2.3 Minor Points

The references are inconsistently formatted, which detracts from the overall presentation. Please ensure that the reference formatting is consistent throughout the manuscript.

The paragraph beginning at line 186 contains mixed fonts. We recommend thoroughly checking the manuscript for any formatting issues before resubmission.

This has now been addressed.

Reviewer 3: SUMMARY OF THE ADVANCE MADE IN THIS PAPER AND ITS POTENTIAL SIGNIFICANCE TO THE FIELD

This manuscript by Kareem and colleagues uses a combination of molecular genetics, including a clonal analysis, and live imaging to analyze the contribution of genes regulated downstream of auxin signaling to the polarities of PIN1 in the shoot apical meristem. Some of the authors have described previously a contribution of the MP signaling effector to regulating PIN1 polarities and they explore here the possibility that a direct target of MP, TMO5, mediates at least some of the effect of MP on PIN1 polarities. The authors address a question essential to understand plant development. However the results do not fully support the conclusions of the authors. One of the main issues is that it takes several days to form organs after induction of sectors with TMO5, suggesting that TMO5 probably only plays a very indirect role in PIN1 polarity. It seems exaggerated to conclude that TMO5 'regulate' PIN1 polarity and it is unclear what we really learn from this work.

SUGGESTIONS TO AUTHORS Main concerns:

3.1 - A general and important concern the data and their presentation. Most of the conclusions are derived from single images (from microscopy or showing phenotypes) in the figures. There is no information about the N for experiments reported in Fig 1, S1, 2, 3 (microscopy images and F,G), S3 (images in C-L), S4. The authors need to provide this information and **extra images** (see below). Without this, their claim are not supported by the data.

We now provide the number of replicates supporting the observations for our confocal data in the figure legends as well as additional images supporting polarity shifts in both Fig. 2K-N and Fig. S2L,M

3.2 Lines 75-94 - Previous publications including one from some of the authors have analyzed the expressions of TMO5 and DOF5.8, and the dependency of their expression on MP. They should be discussed here: Ram et al PLoS genetics 2020, Larriue et al iScience 2022, Mor et al iScience 2022.

We have added these references in the introduction and start of the results in relation to these genes being targets of MP and being expressed in the shoot.

3.3 Lines 96-121 and corresponding figures - In line with comment #1, with single clones being shown on Figures, it is not possible to evaluate the solidity of the results. N and statistics are needed here for all the observations.

We have now included more representative images and quantitative data

3.4 Lines 111 and 116 - The authors mention PIN1 polarity convergence induced by the presence of a TMO5 clone. The PIN1 signal cannot be easily attributed to a specific cells in such images. Can't the authors quantify the polarity to backup their claim? And on Fig 2M,N what is the most obvious is a correlation between high PIN1 and the TMO5 clone. What is the timing of the PIN1 accumulation and possible effect on polarity in relation to the appearance of TMO5? And how does this relate to the organogenesis in term of timing?

We have now included time-lapse images illustrating the timing of PIN1 polarity changes in response to TMO5 activation (Fig. 2K-M). PIN1 polarity was reoriented within 2 days of clone appearance, similar to the effect previously reported for MP (Bhatia et al., 2016). Organ initiation followed within 4-5 days of clone appearance (6-7 days after Dex treatment) (Fig. 2M, O-Q, S2D), indicating that TMO5-induced changes in PIN1 polarity precede and contribute to organogenesis.

3.5 As suggested by Fig 2 and S2, the growth induced by TMO5 clones and effects on PIN1 would take several days. This suggests a largely indirect feedback effect of TMO5 on organogenesis/PIN1 polarities. These experiments do not support the idea of a role 'for TMO5 in controlling PIN1 polarity'. They only suggest that TMO5 can feedback indirectly onto organogenesis/PIN1 polarities. The claim from the author that they have identified a role for TMO5 in the regulation of PIN1

polarity is not supported by the result of their analysis.

The prevailing evidence suggests that auxin itself acts only indirectly in modulating PIN1 polarity, as even in response to MP clones, there is considerable delay before PIN1 is observed to respond (Bhatia et al., 2016). In line with this, the current most favored hypothesis suggests that auxin modifies PIN1 polarity by altering cell wall mechanical properties, which would be expected to take some time. Therefore, we consider the observed timeframes to as expected. From a more cell biology-based perspective, we understand that TMO5 is not involved directly in altering PIN1 polarity. But from a developmental perspective, where the influence of neighboring cells is central, TMO5 activity is very relevant and can be described as a non-cell autonomous modulator of PIN1 polarity. This is further reinforced by the finding that we see changes to PIN1 polarity in response to TMO5 that are unlikely to be associated with organ formation and have included additional data illustrating such cases (see Fig. 3K).

3.6 From line 123-142 - Mor et al iScience 2022 report some shoot meristem defects in TMO5/TMO5-like mutants. This should be discussed.

We have now included several more mentions of the meristem phenotype reported in Mor et al., 2022 in relation to our distinct results on organogenesis.

3.7 Lines 163-184: It is unclear why the authors talk about 'organ outgrowth defects' in the tmo5/t5l quadruple mutants. The mutant can clearly make leaves. Isn't it simply that it grows more slowly? The results of the authors suggest that CK alone is sufficient to accelerate its growth and auxin appears to have a synergistic effect on growth. This would explain why more flowers are observed with auxin+CK on Figure 4. In any case the simple fact that the quadruple mutant can already makes leaves does not allow to conclude that 'cytokinin can substitute for the role of TMO5/T5L ...' (lines 182-184).

In our paper we now acknowledge this is a valid question and relate it to the phenotypes reported by Mor et al. And following this, we again provide evidence that shows that the TMO5Ls contribute to organ formation independently of general shoot growth. This is shown by the "pin" phenotypes and bare vegetative dome of tmo5 single mutants (and worse for multiple mutant combinations) in the presence of low concentrations of NPA. Shoot growth clearly continues since the plant bolts and produces a normal stem yet organ formation is abolished. We have rearranged the order of our results so that the logic should be clearer.

We also see a clear difference in hormone requirement between leaves and flowers. While the tmo5/t5l quadruple mutant can make some leaves, it then terminates. Waiting a longer time does not lead to more leaves being produced. Hence these plants are not simply growing more slowly. Only by providing cytokinin can more leaves form. Hence cytokinin does substitute for the role of TMO5/TFL in promoting later leaf formation. Additional auxin at this stage has no effect, which contrasts with flower formation where both cytokinin and auxin are required. Overall, this indicates that only cytokinin is limiting during vegetative growth while both auxin and cytokinin are limiting during the reproductive phase and is consistent with other published data - for instance in plants multiply mutant for PIN genes, leaves are still produced without additional auxin supplied while flowers are not (Guenot et al., 2012) and we know there is a requirement for both auxin and cytokinin for flower formation (Yoshida et al., 2011).

While we feel the direct contribution of the TMO5Ls to auxin-induced organ growth is well evidenced, we agree that an important question to address in the future is whether the cytokinin contribution acts directly to promote organ growth or whether it more contributes to maintain general meristem function that only indirectly promotes organ growth (or both). On one had we know that cytokinin maintains meristem growth and that the bulk of TCS signal is at the meristem center. On the other hand, there is growing data indicating a direct role for cytokinin in organ formation, e.g. localized TCS activity at primordium positions and the phenotype of ahp6 mutant (Besnard et al., 2014; Kong et al., 2024). The fact that TMO5 and its relatives are expressed in or close to primordium positions adds to this latter line of thinking. We agree that more data, beyond the scope of this paper, would help to conclusively resolve this issue and this uncertainty is now

better articulated in the discussion.

Other concerns:

3.8 Fig3A: the SAM seems abnormally small.

While this meristem is somewhat small, the expression domain appears well defined. The pattern also replicates the expression pattern observed in Mor et al., 2022), which we now more clearly point out in the text.

3.9 Fig 4A-C - a representative situation for the mp would a mutant with 4 leaves (Fig 4C). The authors should put a representative image in Fig 4A.

We have replaced Fig. 4A with a representative image

Resubmission

MS ID#: dev.205255

MS TITLE: TMO5 Regulates PIN1 Polarity convergence and Organogenesis Downstream of MONOPTEROS in the Arabidopsis Shoot

AUTHORS: Marcus G Heisler; Abdul Kareem; Carolyn Ohno

Dear Dr Heisler,

I have now received all the referees reports on the above manuscript, and have reached a decision. The referees' comments are appended below.

The overall evaluation is positive and we would like to publish a revised manuscript in Development, provided that the referees' comments can be satisfactorily addressed. You will see a variety of responses with one of the reviewers requesting additional experiments. If you have these experiments done, feel free to include them. However, it is this editor's judgement that it would be suitable to include a "limitations" section in the text where the concerns (e.g. the timing of TMO5 expression) are laid out.

Otherwise, please attend to all of the reviewers' comments in your revised manuscript. Because many are brief text changes, it's fine to just include these in the highlighted version (rather than going in to detail in a long point-by-point response). However, if you are changing more than a sentence or if you do not agree with any of their criticisms or suggestions explain clearly why this is so in the response to reviewers. If it would be helpful, you are welcome to contact us to discuss your revision in greater detail.

Reviewer 1

SUMMARY OF THE ADVANCE MADE IN THIS PAPER AND ITS POTENTIAL SIGNIFICANCE TO THE FIELD

This manuscript describes a novel role for TMO5/T5L transcription factors downstream of MP in promoting organ (primordium) outgrowth. The work is complete and elegant, and modest and reasonable in its conclusions. While the cellular mechanism of MP-induced organogenesis remains unclear, the authors can at least place TMO5 and TMNO5-triggered CK accumulation in the picture.

SUGGESTIONS TO AUTHORS

I did not review the original submission, and therefore refrain from bringing up new points of attention. I merely evaluate the general soundness of the study and support of conclusions (see

prior section), and to what extent the reviewer's feedback has been implemented. For the latter, all reasonable efforts have been made to address concerns.

Reviewer 2

SUMMARY OF THE ADVANCE MADE IN THIS PAPER AND ITS POTENTIAL SIGNIFICANCE TO THE FIELD

The formation of lateral organs is a fundamental aspect of plant development. Understanding the molecular mechanisms that govern this process provides insight into how plants form and shape their organs in real time. MONOPTEROUS (MP/ARF5) transcription factor plays a pivotal role in organogenesis. Kareem et al. investigated the family of transcription factors known as TARGETS OF MONOPTEROUS (TMO), specifically TMO5, TMO6, and DOF5.8, to identify the downstream components regulated by MP. Their study built upon previous findings by Bhatia et al. (2016), which demonstrated that MP/ARF5 regulates the polar localization of the PIN1 protein in a non-cell-autonomous manner. This regulation is significant because auxin, a key plant hormone, controls the expression of PIN1, thereby influencing its own movement within the plant. The polarization of PIN1 is a crucial event for lateral organ development. The polar localization of the PIN1 protein ensures the accumulation of auxin in specific regions within the meristem, which is essential for cells in those areas to adopt the fate of a primordium - the initial stage of organ formation.

Through genetic analysis, Kareem et al. demonstrated that among the TMO family members, the expression of TMO5 is dependent on a functional MP transcription factor. Furthermore, the authors showed - using clonal induction experiments - that TMO5 alone can promote the polar localization of PIN1 even in the absence of MP. This finding suggests that TMO5 can initiate organ formation in the shoot apical meristem independently.

When the quadruple mutants of *tmo5/t5ls* were identified, it was revealed that these genes were indeed involved in the process of organogenesis. When a combination of auxin and cytokinin hormones was applied to these quadruple mutant plants, the researchers observed the formation of lateral organs. This result suggests that the TMO5/T5L genes play a crucial role in both auxin and cytokinin signaling pathways that regulate lateral organ development.

SUGGESTIONS TO AUTHORS

The study has been carefully designed and supported with relevant data. However, the discrepancy in the timing of MP and TMO5 expression in the incipient primordium raises a question related to this regulation. In an earlier study (Bhatia et al, 2016), the group showed that PIN1 polarization follows MP expression. The positive feedback governed by auxin signaling is the cornerstone for robust organogenesis in the plant meristem. Similarly, Heisler et al. (2005) demonstrated that PIN1-GFP is strongly polarly localized in I1, and the P1 positions where TMO5-GFP expression is initially observed largely exhibit reversals in PIN1 polarity.

This discrepancy observed in the expression of MP and TMO5 is perhaps due to the use of translational fusions. There is a delay in the maturation of GFP/YFP, and in addition, a sensitivity issue arises in detecting fluorescent proteins. This could be resolved by carrying out in situ hybridization against the MP and TMO5 transcripts in the meristem.

Reviewer 3

SUMMARY OF THE ADVANCE MADE IN THIS PAPER AND ITS POTENTIAL SIGNIFICANCE TO THE FIELD

Karim et al show that the previously-identified MONOPTEROS (MP) target, TARGET OF MONOPTEROS 5 (TMO5) together with 3 related genes are expressed in the SAM, mainly subepidermally, and are required for shoot growth and organogenesis. They demonstrate that TMO5 clonal sectors result in elevated PIN1 expression and reorient PIN1 polarity non-cell autonomously. Finally, they show that the organogenesis defects displayed in both *mp* and *tmo5* *tmo5*-like mutants can be partially rescued by exogenous application of auxin and cytokinin.

Overall, the manuscript represents a significant contribution to the field of plant development. It was previously found, through studying genetic mosaics, that MP is sufficient to re orient PIN1 polarity and this study indicates that the MP target TMO5 has a similar effect thus advancing our understanding on what causes PIN1 reorientation, which is a fundamental problem in plant development. The data presented is overall high quality and the authors conduct careful important work. The mosaics are strong point and only few labs conduct these kind of experiments at the level presented here.

SUGGESTIONS TO AUTHORS

The exogenous hormone application work is intriguing and appropriate for publication however no genetic evidence is provided to explore/support the proposed interactions. We propose that this limitation is discussed in the text, highlighting that this point can be addressed in future work.

The comments are meant to increase accuracy/improve impact of what is already a very nice study.

Title: "Vascular" really needed? The TMO5 characterization shown here indicates that TMO5 is also epidermal?

Introduction. Line 48: "likely by altering mechanical stresses". Given that this is not demonstrated experimentally, perhaps by altering..." seems more appropriate.

Fig. 1A: Is there a mistake in the order of the primordia labels?

Sectors.

We could not find an explanation of why two different mp alleles are used in the experiments. If this is not described in the manuscript, this information could be added where the authors consider most appropriate-perhaps the methods.

The authors state that, unlike TMO5, DOF5.8 and TMO6 clones are not sufficient to promote PIN1 expression nor to trigger outgrowths. However, PIN1 repolarization is not mentioned-is there a reason for this please clarify.

Figure 2L: The cells with arrows indicating PIN1 polarity don't all seem to have a clear signal crescent to discriminate the origin of the signal between two abutting cell membranes (or at least it is not possible to appreciate from the projection shown in the figure). In general, cells within TMO5 clones have high PIN1 expression compared to the neighboring cells. This difference, together with the inability of confocal microscopy to resolve adjacent membranes, may lead to wrongly assuming that the adjacent cells polarize PIN1 towards the clone. Maybe it would be safer to remove ambiguous arrows? A few arrows in panels M-P may be also be affected by this issue. It would be good to look at raw data again in 3D on a cell by cell basis to help clarify this point.

To assess the relevance of TMO5 on epidermal PIN1 polarity patterning: Is the time elapsed between the appearance of TMO5 clones and PIN1 repolarization compatible with the flower plastochron length. This information might help with understanding the connection between repolarization and organ initiation.

tmo5 tmo5-like mutants:

Each TMO5-LIKE gene studied seems to have a unique expression pattern, TMO5-LIKE4 appearing complementary to TMO5. The authors show that several combinations of tmo5 tmo-like produce less siliques and their phyllotaxis is abnormal. To understand this phenotype, it would be good to analyze a PIN1 reporter in these mutant backgrounds. This could also enable a better understanding

of how auxin+CK treatment can partially rescue flower initiation in the *tmo tmo5*-like quadruple mutant. Do the authors have such data? If not this can perhaps be highlighted as potential future work.

Mutant rescue with auxin+ck treatment:

Figure 3A-D: In panel A the two channels don not look aligned. In panel C, unlike in the other panels, there is no PIN1-CFP channel shown. Is this a mistake or simply this line is not available?

Lines 185, 189, 197: The trans-zeatin concentrations used in different experiments are very different. Clarify in methods why that is?

Figure 4D, variable of the y axis and lines 186-192: It is not obvious what exactly was measured here and what is the meaning of "Outgrowth (%)". Please clarify and maybe show the datapoints.

Conclusion paragraph:

Line 216: "Our data suggest that [...] organogenesis involves an amplification process [...] via the induction of TMO5-mediated auxin synthesis or signaling". Is this the only possible interpretation? Could TMO5 be acting primarily on auxin transport by regulating PIN1 expression and activity? Or by promoting expression of auxin importers?

Please consider the point mentioned in the start of the comments that the exogenous application experiments are not as definitive as the genetic mosaics particularly as high concentrations of hormones are used so results need to be appropriately qualified.

First revision

Author response to reviewers' comments

REVIEWER 2 SUGGESTIONS TO AUTHORS

The study has been carefully designed and supported with relevant data. However, the discrepancy in the timing of MP and TMO5 expression in the incipient primordium raises a question related to this regulation. In an earlier study (Bhatia et al, 2016), the group showed that PIN1 polarization follows MP expression. The positive feedback governed by auxin signaling is the cornerstone for robust organogenesis in the plant meristem. Similarly, Heisler et al. (2005) demonstrated that PIN1-GFP is strongly polarly localized in I1, and the P1 positions where TMO5-GFP expression is initially observed largely exhibit reversals in PIN1 polarity.

This discrepancy observed in the expression of MP and TMO5 is perhaps due to the use of translational fusions. There is a delay in the maturation of GFP/YFP, and in addition, a sensitivity issue arises in detecting fluorescent proteins. This could be resolved by carrying out in situ hybridization against the MP and TMO5 transcripts in the meristem.

Our apologies, the primordia were mis-labelled. This has now been corrected. Early TMO5 expression occurs at I1 when polarity convergence is strengthening. More TMO5 expression is apparent at P1, P2 when outgrowth accelerates. There is still a convergence at the organ tip, despite reversal adaxial to the primordium, consistent with the response of PIN1 polarity to TMO5 clones.

REVIEWER 3 SUGGESTIONS TO AUTHORS

The exogenous hormone application work is intriguing and appropriate for publication however no genetic evidence is provided to explore/support the proposed interactions. We propose that **this limitation is discussed in the text**, highlighting that this point can be addressed in future work.

We have modified the text of our final sentence on future work, explicitly mentioning the identification of the genes involved.

The comments are meant to increase accuracy/improve impact of what is already a very nice study.

Title: "Vascular" really needed? The TMO5 characterization shown here indicates that TMO5 is also epidermal?

We have shortened the title accordingly.

Introduction. Line 48: "likely by altering mechanical stresses". Given that this is not demonstrated experimentally, perhaps by altering..." seems more appropriate.

Done

Fig. 1A: Is there a mistake in the order of the primordia labels?

See comments above.

Sectors.

We could not find an explanation of why two different *mp* alleles are used in the experiments. If this is not described in the manuscript, this information could be added where the authors consider most appropriate-perhaps the methods.

We have added to the methods: "The *mp-T370* mutant allele is in the Ler ecotype and *mp-B4149* allele is in the Utrecht ecotype (Bhatia et al., 2016; Hardtke and Berleth, 1998; Weijers et al., 2006). Both *mp* alleles were used to test for ecotype-specific effects."

We have also added in the results section: "Towards this end, we generated small clones of cells expressing these genes individually in the *mp* mutant shoot meristem (*mp-T370* and *mp-B4149*, both strong alleles used to exclude ecotype-specific effects) harbouring a PIN1-CFP (or PIN1-GFP) marker."

The authors state that, unlike TMO5, DOF5.8 and TMO6 clones are not sufficient to promote PIN1 expression nor to trigger outgrowths. However, PIN1 repolarization is not mentioned-is there a reason for this please clarify.

The polarity was not clear in both the cases.

Figure 2L: The cells with arrows indicating PIN1 polarity don't all seem to have a clear signal crescent to discriminate the origin of the signal between two abutting cell membranes (or at least it is not possible to appreciate from the projection shown in the figure). In general, cells within TMO5 clones have high PIN1 expression compared to the neighboring cells. This difference, together with the inability of confocal microscopy to resolve adjacent membranes, may lead to wrongly assuming that the adjacent cells polarize PIN1 towards the clone. Maybe it would be safer to remove ambiguous arrows? A few arrows in panels M-P may be also be affected by this issue. It would be good to look at raw data again in 3D on a cell by cell basis to help clarify this point.

We have removed the arrows we think were ambiguous.

To assess the relevance of TMO5 on epidermal PIN1 polarity patterning: Is the time elapsed between the appearance of TMO5 clones and PIN1 repolarization compatible with the flower plastochron length. This information might help with understanding the connection between repolarization and organ initiation.

Firstly, during wild type development, *TMO5* and related genes are likely not acting alone to promote PIN1 polarity convergence and organ growth. This would explain why changes in PIN1 expression, polarity and tissue growth in response to *TMO5* clones may be delayed after experimental induction compared to how quickly corresponding changes follow the appearance of *TMO5* during wild type development.

The flower plastochron length is around 24 hrs. We first detected PIN1 convergence in response to clones around 24 hrs after clone appearance. So, if *TMO5* expression initiates during I1 (a 24 hr period), this is consistent with early *TMO5* expression contributing to the later strengthening of the PIN1 convergence during that I1 stage. During wild type development, organ outgrowth accelerates during P1, 48 hours after the first appearance of *TMO5*, which is faster than observed in response to *TMO5* clones (4 days).

tmo5 tmo5-like mutants:

Each *TMO5*-LIKE gene studied seems to have a unique expression pattern, *TMO5*-LIKE4 appearing complementary to *TMO5*. The authors show that several combinations of *tmo5* *tmo5*-like produce less siliques and their phyllotaxis is abnormal. To understand this phenotype, it would be good to analyze a PIN1 reporter in these mutant backgrounds. This could also enable a better understanding of how auxin+CK treatment can partially rescue flower initiation in the *tmo5* *tmo5*-like quadruple mutant. Do the authors have such data? If not this can perhaps be highlighted as potential future work.

We have edited the summary sentence to acknowledge this point. E.g.

“These data demonstrate that despite the noted differences in expression, *TMO5* and the *T5L* genes contribute redundantly to organogenesis independently of shoot growth, in synergy with auxin and its polar transport.”.

Mutant rescue with auxin+ck treatment:

Figure 3A-D: In panel A the two channels don not look aligned. In panel C, unlike in the other panels, there is no PIN1-CFP channel shown. Is this a mistake or simply this line is not available?

There are some places where *T5L1* and *T5L4* looks outside the PIN1-GFP marked tissue. This is because of strong expression of the *T5Ls* from optical sections near the bottom of the stack of images projected. We have now removed these lower optical sections to make the figure clearer. For *T5L3::n3GFP*, we simply did not have PIN1-GFP in this line.

Lines 185, 189, 197: The trans-zeatin concentrations used in different experiments are very different. Clarify in methods why that is?

While we found that 10uM TZ was sufficient to induce leaves in *mp* dome meristem, a higher concentration of TZ (1mM) was used to treat the pin-like inflorescence, similar to the concentration used in Yoshida et al (2011), as 10uM did not show a clear effect. To make this clearer we have modified the sentence describing this in the results section as follows:

“Instead, we found that a combined application of auxin (5mM Indole-3-acetic acid (IAA)) and a concentration of cytokinin (1mM TZ) previously shown to induce outgrowth on dark-grown pin-like inflorescences in tomato (Yoshida et al., 2011), was sufficient to promote organ primordia-like outgrowth (Fig. 4D)”

Figure 4D, variable of the y axis and lines 186-192: It is not obvious what exactly was measured here and what is the meaning of "Outgrowth (%)". Please clarify and maybe show the datapoints.

The figure legend has been modified to better describe the data. It now reads “Frequency of *mp* pin-like inflorescence meristems exhibiting organ primordia-like outgrowths after treatment with

both...". Also, Fig. 4D Y-axis label has been edited to read 'PIN-like inflorescences with organ primordia-like outgrowths (%)' .

Conclusion paragraph:

Line 216: "Our data suggest that [...] organogenesis involves an amplification process [...] via the induction of TMO5-mediated auxin synthesis or signaling". Is this the only possible interpretation? Could TMO5 be acting primarily on auxin transport by regulating PIN1 expression and activity? Or by promoting expression of auxin importers?

We have modified the sentence to reflect the additional possibilities suggested.

Please consider the point mentioned in the start of the comments that the exogenous application experiments are not as definitive as the genetic mosaics particularly as high concentrations of hormones are used so results need to be appropriately qualified.

Second decision letter

MS ID#: dev.205255R1

MS TITLE: TMO5 Regulates PIN1 Polarity convergence and Organogenesis Downstream of MONOPTEROS in the Arabidopsis Shoot

AUTHORS: Marcus G Heisler; Abdul Kareem; Carolyn Ohno

Dear Dr Heisler,

I am happy to tell you that your manuscript has been accepted for publication in Development, pending our standard publication integrity checks.